



# The impact of initial conditions on convection-permitting simulations of flood events

Lu Li[1], Marie Pontoppidan[1], Stefan Sobolowski[1], Alfonso Senatore[2]

[1]NORCE Norwegian Research Centre, Bjerknes Centre for Climate Research, Jahnebakken 5, 5008 Bergen, Norway
[2]Department of Environmental and Chemical Engineering, University of Calabria, Italy

*Correspondence to*: Lu Li (luli@norceresearch.no)

**Abstract.** Western Norway suffered major flooding after 4 days of intense rainfall during the last week of October 2014, resulting in damages totalling hundreds of millions Norwegian kroner. These types of events are expected to become more frequent and severe as Norwegian (and the earth's) climate continues to warm. However, due to the strong effects that local
features and conditions can have on these kinds of events, coarse-grained global or regional models are unable to capture their characteristics. Very high resolution models run at so-called convection-permitting scales have shown some promise for reliably capturing such events. Doing so in a robust manner is, as a matter of course, of high interest to both scientists and stakeholders in both climate prediction and projection contexts. Despite this promise, the impacts of initial conditions on convection-permitting simulations, i.e., precipitation pattern and discharge, are uncertain, especially over complex,
mountainous terrain. Complicating matters, these areas also usually lack dense measurement networks. In this paper, we apply a distributed dynamic regional atmosphere-hydrological modelling system (WRF-Hydro) at convection-permitting scale and assess its performance over four catchments in western Norway for the aforementioned flood event. The model is calibrated and then evaluated using observations and benchmarks obtained from the HBV light model. Interestingly, the calibrated WRF-Hydro model with NSE value of 0.86 exceeds the upper benchmark obtained from the conceptual HBV
model with NSE value of 0.80 suggesting that the former performs as well or better than the simpler conceptual model, especially in areas with complex terrain and poor observational coverage. Confident in the capabilities of the modelling system we then examined the sensitivity of precipitation pattern and discharge, especially peak flow, to poorly constrained elements such as spinup time and snow conditions. The results show that: (1) overall the convection-permitting WRF-Hydro simulation captures the precipitation pattern/amount, the peak flow volume and the timing of the flood event; (2)
precipitation is not overly sensitive to spinup time, while discharge is slightly more sensitive due to the influence of soil moisture, especially during the pre-peak phase; (3) the idealized snow depth experiments show that a maximum of 0.5 m of snow is converted to runoff irrespective of the initial snow depth and that this snowmelt contributes to discharge mostly during the rainy and the peak flow periods. This suggests that snow-cover, in these experiment at least, intensifies the extreme discharge instead of acting as sponge, which further implies that future rain-on-snow events may contribute to





higher flood risk. While targeted experiments on the changing characteristics of projected future rain-on-snow events are needed to confirm this study suggests that WRF-Hydro is an ideally formulated tool to investigate these questions.

## 1 Introduction

Heavy rainfall, with local amounts exceeding 350 mm, fell over the coastal and mountainous areas of western Norway between 26 - 29 October 2014. The event caused widespread flooding, with 16 stations registering discharge above the 50-

year flood threshold (Langsholt et al., 2015). The severe flooding of many regional river systems destroyed infrastructure, houses and isolated towns. Overall the damage exceeded 131.6 million Norwegian kroner (Dannevig et al., 2016). A list of reports (in Norwegian), e.g. "October flood in western Norway 2014" (Dannevig et al., 2016) and "The flood in western Norway October 2014" (Lansholt et al., 2015) from Norwegian Water Resources and Energy Directorate (NVE) were produced, which documented the rainfall and discharge records and societal impacts. For flood hazards such as this, it is a

challenge to forecast/hindcast the hydrological response due to the complex terrain and the events' complex spatial and temporal characteristics. However, with extreme precipitation over this region projected to increase significantly over the coming decades (e.g. Hanssen-Bauer et al., 2017) the need to reliably reproduce such events is high for both climate researchers and operational professionals.

### 1.1 Norwegian flood types, changes & rain-on-snow

The complex and varying terrain of Norway divides the country into different climatic zones and flood-generating processes vary by regions. For example, northern and eastern Norway are mainly prone to spring snowmelt floods while southern and western Norway are dominated by rain induced floods (Vormoor et al., 2015). According to recent studies, snowmelt generated floods have decreased and shifted earlier in the spring in recent decades (Vormoor et al., 2016; Pall et al. 2018). At the same time, rain dominated floods are increasing in frequency (Vormoor et al., 2016). This is consistent with observed

increases in precipitation (Dyrrdal et al., 2012) and streamflow (Stahl et al., 2010; Wilson et al., 2010), and the trends are projected to continue into the future (Hanssen-Bauer et al., 2017; Sorteberg et al., 2018). Though temperatures are increasing, much of the winter precipitation in inland catchments will continue to fall as snow until at least the mid-century (Hanssen-Bauer et al., 2017). However, as temperature rises many of these catchments may experience more rain-on-snow events. Warm, and often windy conditions during such events can cause substantial additional snowmelt, which can

exacerbate an already dangerous flooding event (Marks et al., 1998, 2001). In fact, for many catchments in the world, such as the western US, rain-on-snow events are important for prediction of flood responses and risk (Berghuijs et al., 2016; Musselman et al., 2018). While earlier snowmelt is decreasing the frequency of such events in the late spring and at low altitudes, both the magnitude and frequency of rain-on-snow events are increasing during winter in central Europe (Freudiger


et al., 2014), and likely also in Norway. Pall et al. (2018) used a high-resolution (1 km) seNorge data set to construct

climatology of rain-on-snow occurrence in the mainland Norway for recent decades. They found an increase in rain-on-snow events in high-elevation areas across the mainland in winter-spring. Given the dependence of floods in Norway on a complex interplay between variations in elevation, temperature gradients (e.g. between land and ocean), orographic interactions, existing snow and soil moisture distributions, etc. it is critical to run models (either dynamical or statistical) at resolutions that can capture this complexity. But this requirement presents challenges of its own. In addition, few studies have assessed

the causality of flood occurrence (extreme streamflow) resulting from rain-on-snow signals at the local hydrological scale (e.g. Surfleet and Tullos, 2013; Corripio and López-Moreno 2017). Despite this, the rain versus snowmelt contribution to the flow can be important in determining the flood generation processes for Norway and can be particularly sensitive to the vertical temperature gradient.

### 1.2 Forcing data and convection-permitting modelling

In order to improve our scientific understanding as well as predictions and projections of flooding, high quality meteorological forcing data is crucial. A lack of detailed precipitation records that accurately represent spatial and temporal variability at both the basin and regional scales presents well-known challenges to hydrological modelling.. In mountainous areas, like western Norway, where precipitation is strongly influenced by the terrain, spatial patterns of precipitation are not well captured by either sparse gauge data or gridded precipitation datasets (e.g. satellite-based products or high-resolution

interpolation-based datasets). High-resolution, convection-permitting modelling has exhibited great promise in addressing these issues and has potential as a powerful tool for hydrological prediction (Prein et al., 2015, 2016, 2017; Smiatek et al., 2016; Kendon et al., 2017; El-Samra et al., 2018; Poschlod et al., 2018; Avolio et al., 2019). Pontoppidan et al. (2017) investigated the atmospheric conditions leading to the aforementioned flooding event in October 2014 over Western Norway and found that convection permitting simulations (~3km grid spacing) from the Weather Research and Forecasting (WRF)

model substantially improved the representation of precipitation compared to a coarser resolution reanalysis. This improvement was seen both in terms of absolute values and spatial-temporal distribution. The largest improvement was found with the resolution jump from parameterized to explicitly resolved convection (e.g., 9km to 3km over western Norway). Several previous studies over other regions also demonstrate the added-value of convection permitting modelling for extreme weather impact studies in regions with complex terrain. For example, Maussion et al. (2011) showed improved

representation of precipitation in a convection-permitting (2km) simulation when compared to satellite products over the Himalayan region. El-Samra et al. (2018) suggest that downscaling over complex terrain requires a horizontal grid resolution of 3km or higher in to improve the forecasting of mean and extreme temperatures and capture the orographic precipitation climatology. Conversely, coarse resolution (~ 9km) simulations miss the impact of orography on temperature and precipitation. Additionally, the studies of Rasmussen et al. (2011, 2014) found that a spatial and temporal depiction of



snowfall that is adequate for water resource management over the Colorado Headwaters regions can only be achieved with
the appropriate choice of model grid spacing and parameterizations. The modeling systems that are capable of accurately
depicting the atmosphere at these scales now increasingly incorporate other regional system components such as crops,
urban features and, of most relevance to the present study, hydrology.

**1.3 A dynamical hydrometeorological model: WRF-Hydro modelling system**

The Weather Research and Forecasting Model Hydrological modeling system (WRF-Hydro) is a model coupling framework
designed to link multi-scale process models of the atmosphere and terrestrial hydrology (Gochis et al., 2018). It runs both in
fully-coupled (two-way) or uncoupled (one-way, from atmosphere to land) modes and is intended to serve as both a
hydrometeorological prediction system and a research tool. The system has been applied in studies around the world (e.g.
Senatore et al. 2015; Givati et al. 2016; Arnault et al., 2016; Xiang et al., 2017; Naabil et al., 2017; Verri et al., 2017; Lin et
al., 2018; Rummler et al., 2019).  It is currently in use operationally as a key component of the United States national water
model where it expands the number streamflow forecast points from ~3600 points to ~2.7 million river reaches
(https://water.noaa.gov/about/nwm). WRF-Hydro has also been applied in Africa (Arnault et al., 2016; Kerandi et al., 2018),
in the Himalayas (Li et al., 2017), in Italy (Verri et al., 2017; Senatore et al., 2015) and in Eastern Alps (Rummler et al.,
2019) with promising results and shows potential for use in runoff forecasting, water resource planning and climate changes
impact assessments. However, despite application across a diverse array of catchments and research questions, the system
has yet to be evaluated for a case in Norway.

There are still challenges to discharge prediction by WRF-Hydro and the performance varies across geographic regions and
climate. For example, it simulated flood events in the Black Sea region fairly well if both model calibration and WRF data
assimilation were performed jointly, while the streamflow obtained with raw WRF precipitation was in general very poor
(Yucel et al., 2015). It also simulated a full annual cycle of the Crati River basin in southern Italy with Nash-Sutcliffe
Efficiency (NSE) of 0.8 using observed precipitation while only achieved an NSE of 0.27 using simulated precipitation
(Senatore et al., 2015). Naabil et al. (2017) applied WRF-Hydro in a test case over west Africa for water resources
management, and found that further improvements via proper model calibration and consideration of the effects of model
biases in dam level were recommended, although model captured the attributes of the streamflow. Furthermore, Verri et al.
(2017) demonstrated that the performance of WRF-Hydro was severely affected by various components including simulated
precipitation, initial conditions and also the calibration / validation of discharge hydrography. In Texas, WRF-Hydro has
shown promise as a forecast tool, but suffers from poor prediction skill in areas with human altered flows, in which both the
surface runoff and the base flow are underpredicted (Lin et al., 2018). Additional studies also noted the sensitivity of WRF-
Hydro to the initial conditions (spinup time) (Roman-Cascon et al., 2016; Bonekamp et al., 2018; Verri et al., 2017). In order
to obtain a stable WRF-Hydro simulation, a spinup period is required, which depends on the quality of the model input and



soil data. Therefore, the impact of the spinup time needs to be assessed on per-case basis as it likely depends on local conditions.

### 1.4 Objectives of the paper

Due to the traditional separation of hydrological and atmospheric modelling communities, significant gaps exist in our
knowledge of the full-chain responses to hydrometeorological extremes, from circulation/transport of moisture to precipitation to discharge. WRF-Hydro is designed to link across these components and their characteristic scales to provide a modelling framework that can address these gaps (Gochis et al., 2018).

In this study, we employ WRF-Hydro in western Norway to investigate meteorological and hydrological processes driving the October 2014 flooding. To our knowledge, this is the first study using a complete meteorological-hydrological modelling
approach to characterize a precipitation induced extreme flooding event in Norway. The causal mechanisms and evolution of this particular flood event are examined. In addition, we explore the sensitivity of the discharge to different initial conditions such as soil and snow.

The work uses the output of Pontoppidan et al. (2017) as its starting point for the simulation of the meteorological processes and the hydrological impact. As such, an offline configuration for the WRF-Hydro model is chosen. This is because we
primarily aim to understand the flood event in the context of its hydrological response to the weather forcing and land surface conditions. Feedbacks between the atmosphere and land, though important, generate second order effects that likely have only small impact within such a short duration event. Also, the offline mode of WRF-Hydro system is preferable for our study as it provides a clearer interpretation of the results, identification of uncertainties in the water budget and assessment of sensitivity to critical parameters in the atmospheric and hydrological components (Li et al., 2017).

The remainder of the paper is structured as follows: after the introduction, a description of the study area and data is presented, followed by methods, including a description of the WRF-Hydro setup and experiment design, model calibration and benchmark evaluation. Results concentrate on the model calibration and benchmark evaluation, precipitation evaluation, and the impacts of initialization (spinup time) and prescribed snow cover are examined. Finally, the main conclusions are presented.

## 2 Study area and data

### 2.1 Study area

Our four study catchments are located in western Norway, where the landscape is dominated by steep orography and complex terrain due to the fjords and elevation varies from sea level to more than 2400m (Figure 1). The complex terrain both enhances the precipitation and generates large local differences in the precipitation distribution (e.g. Reuder et al. 2007,



Pontoppidan et al 2017). Norway is positioned at the exit region of the North Atlantic storm track, which brings low-pressure systems and associated frontal precipitation towards the west coast on a regular basis during autumn and winter. Western Norway is the wettest part of the country (Hanssen-Bauer and Førland, 2000) and annual precipitation exceeds 3000 mm in several places; but there is also high spatial variability. For example, Kvamskogen-Jonshøgdi (60.389 N and 5.964 E) records 3151 mm while Vossevangen (60.625 N and 6.426 E), which is only 36.6 km away, only receives 1280 mm (MET

Norway, 2015) (see in Figure 1).

### 2.2 Hydro-meteorological conditions

October 2014 was wetter than usual in western Norway. The situation was maintained by an atmospheric river and the passage of multiple frontal systems with moderate to heavy precipitation. Two days before the flooding event, on the 26th of October, a low-pressure system with associated fronts passed over western Norway and delivered considerable amounts of

precipitation. A cold front passed the area at midnight on the 27th, advecting colder and drier air into the area for a short period. Simultaneously, a disturbance over Scotland developed and moved towards Norway, leaving western Norway in the warm sector of an intensifying low-pressure system. Once again large amounts of precipitation fell from midday on the 27th to early evening on the 28th. The associated cold front passed the Bergen area in the afternoon and the precipitation intensity decreased with its passage. Due to the already saturated soil and several days with more or less continuous rainfall, the flood

peaked in the Voss area in early evening of the 28th (Pontoppidan et al., 2017).

According to the NVE report made by Langsholt et al. (2015), there was shallow snow cover in high-altitude areas east of the catchments. In Voss however, where our four catchments are located, there was no snow in the snow depth water equivalent maps released by NVE. These maps are made from model simulations based on snow observations from NVE. Discharge in each of the study catchments was over the 50-year return level. On the 29th of October, the daily discharge

record held since 1892 was broken at the Bulken station, located at the outlet of Vangsvatnet (Langsholt et al., 2015).

### 2.3 Observational data

We use 43 precipitation gauges from the Norwegian Meteorological Institute (MET Norway) situated in and around the catchments, with either hourly or daily precipitation data. Typically rain gauges in Norway are deployed at low elevations and in valleys resulting in skewed precipitation distributions. To rectify this, 11 HOBO rain gauges, which provide hourly

data, were deployed at higher elevations in a transect from the coast to inland (Pontoppidan et al., 2017). A table with station details can be found in Pontoppidan et al. (2017). Catchment averaged precipitation is calculated as a mean of the rain gauges positioned within the affected catchment. Four discharge stations from NVE are used for WRF-Hydro model discharge calibration and validation (see Table 1). It should be noted that the drainage basin of the Bulken catchment



includes the catchments of Kinne and Myrkdalsvatn. Figure 1 shows the locations of the four catchments and the
measurement sites of rainfall and discharge gauges.

## 3 Methods

### 3.1 WRF domain design

The Advanced Research WRF (WRF-ARW) model version 3.9.1 is set up with two nested domains with spatial resolution of
9 km and 3 km (Figure 1). The lateral boundaries are forced with the 6-hourly ERA-interim reanalysis with a spatial
resolution of 0.75 degrees (Dee et al. 2011). The sea surface temperatures (SST) are also updated every 6 hours. The model
is run with 40 vertical levels in all domains.

The choice of the microphysical scheme is important for precipitation. Previous studies of mountain precipitation using
WRF have shown that the Thompson microphysical scheme (Thompson et al. 2008) performs well (Collier et al. 2013;
Maussion et al. 2014; Rasmussen et al. 2011, 2014, Li et al. 2017), especially in areas with mixed hydrometeors because it
computes cloud water, rain water, snow, graupel and ice. The scheme was also successfully used in a previous study on this
specific event (Pontoppidan et al. 2018). The grid spacing in the outer domain is in the so-called "gray zone" (5-10 km)
where convection may or may not be explicitly resolved; therefore, we tested the impact of the convection parameterization
on precipitation. The results showed negligible differences between simulations with the convection scheme on and off. Here
we present the results from the simulations with the convection parameterization turned off. The Yonsei University scheme
(Hong et al., 2006) is used for the planetary boundary layer, the RRTM scheme for long wave radiation (Mlawer et al., 1997)
and the RRTMG scheme for shortwave radiation (Iacono et al., 2008). The Noah Land Surface model ('Noah LSM',
Mitchell et al. 2001) is used for surface scheme, which has a bulk layer simple canopy and snow model. In the Noah LSM,
the snow cover area fraction within a model grid is determined as a function of snow water equivalent (SWE) using a
generalized snow depletion curve. When snow is on the ground, the model considers a bulk snow-soil-canopy layer and
computes surface temperature at each time step. The snow surface temperature for the snowpack is estimated in two steps.
Firstly, the energy balance between the snowpack, top soil layer and the overlying air is calculated to obtain an intermediate
temperature. This temperature can rise above freezing even when the model grid is fully covered with snow. Secondly, the
effective temperature is adjusted by accounting for the fractional snow cover on the ground (Livneh et al., 2010).

Additional configuration details include a model time-step of 18 seconds over the inner domain and the use of spectral
nudging to keep the large-scale flow consistent with the driving ERA-Interim reanalysis. This approach proved to be useful
when reproducing extreme precipitation events due to the better resolved synoptic scale features over North Atlantic
(Heikkilä et al., 2011). Spectral nudging (Radu et al., 2008) is only applied in the outer domain leaving the model free to



create its own structures in the inner domain. In the present case, nudging is only applied above the boundary layer and only on wavelengths longer than 585 km.

### 3.2 WRF-Hydro modelling system

Version 3.0 of the WRF-Hydro modelling system is used in the study. A comprehensive description of the model system can be found in Gochis et al. (2015). In our study, the saturated subsurface overflow routing, surface overland flow routing, channel routing and base-flow modules are activated. The overland flow routing adopts a 2-D diffusive wave formulation (Julien et al., 1995) and the channel routing is calculated by a 1-D variable time-stepping diffusive wave formulation. In addition, a bucket model for base-flow is used where a groundwater reservoir with a conceptual depth and a related conceptual volume is associated. A few lakes in Bulken catchment are not considered in this study due to lack of data.

WRF-Hydro is set up to run offline using the WRF atmospheric simulations as input (see Introduction). The subgrid routing processes are executed at a 300m grid spacing and the surface physiographic files are prepared by ARCGIS 10.6 (Sampson et al. 2018). The physiographic file includes high-resolution terrain grids specifying the topography, channel grid, flow direction, stream order (for channel routing), ground-water basin mask and the position of stream gauging stations which are the outlets for water routing out across the landscape (Gochis et al., 2015). There are four stream orders in the network of the study catchments shown in Figure 2.

### 3.3 Model calibration

Two-step calibration of WRF-Hydro is performed in the study. First, we select the most sensitive parameters from a wide range of parameters. These include: the saturation soil conductivity (in SOILPARM.TBL), optimum transpiration air temperature (in VEGPARM.TBL) and infiltration parameter (in the surface runoff parameterization of GENPARM.TBL), Manning roughness coefficients (in the channel routing of CHANPARM.TBL), the groundwater bucket model exponent (in the ground water bucket model of GWBUCKPARM.TBL), surface flow roughness scaling factor (OVROUGHRTFAC) and the surface retention depth (RETDEPRT) (Yucel et al., 2015; Li et al., 2017). Second, three parameters, which are particularly sensitive, are tuned using the auto-calibration Parameter Estimation Tool (PEST http://www.pesthomepage.org): two infiltration parameters, i.e. REFDK_DATA (refdk) and REFKDT_DATA (refkdt), which are important for surface runoff, and the Manning routing coefficients (mn01). The offline model is then forced by meteorological output data and calibrated based on the observed discharge in the Svartavatn catchment. The remaining three catchments (i.e. Bulken, Kinne and Myrkdalsvatn) are used for validation and evaluation of the parameters' transferability. The simulations are initialized on 1 September 2014 and run until 1 November 2014. In order to remove the impact of initialization we use the first 30 days as spinup in the model calibration. The best parameter set is then chosen based on the Nash-Sutcliffe efficiency (NSE)



coefficient (Nash and Sutcliffe, 1970). Two more indices, bias and root mean square error (RMSE) are also used for validation. A perfect model would have an NSE value of 1 and bias and RMSE values equal to 0.

### 3.4 Benchmark evaluation approach

A simple bucket-type Hydrologiska Byråns Vattenbalansavdelning (HBV) light model was used as a benchmark for model comparison and evaluation (Seibert and Vis, 2012; Seibert et al., 2018). The HBV light is an offshoot of the HBV model developed in the 1970s by the Swedish Meteorological and Hydrological Institute (SMHI). It consists of four main routines, i.e., snow-, soil-, routing- and response routine and simulates daily discharge using daily precipitation, temperature and potential evapotranspiration (Seibert and Vis, 2012). Its strength lies in the relatively low requirements for input data and the

limited number of parameters (Rusli et al., 2015). Here, the calibrated HBV streamflow is used as upper benchmark ($R_{upper}$) and two alternatives are then used as lower benchmark ($R_{lower}$), one generated from the mean streamflow from 1000 random parameter sets ($R_{lower/random}$) and another from the regionalization parameter set from other nearby catchments ($R_{lower/regional}$). The catchment averaged daily precipitation, temperature and potential evaporation from WRF are used as input in the HBV model simulation. To maintain consistency with the WRF-Hydro modeling, the HBV simulations are also initiated on 1

September 2014 and run until 1 November 2014 with the first 30 days used as spinup. The performance measure for the benchmark evaluation is the NSE.

### 3.5 Initialisation experiments

Previous studies found that spinup time influences the initial conditions such as the soil moisture content and therefore the latent heat flux, which in turn influences the precipitation (Kleczek et al., 2014; Bonekamp et al., 2018; Verri et al., 2017).

Jankov et al. (2007) suggested the spinup time should be at least 12 h to prevent instabilities in WRF, but the recommended length most likely depends on the input quality and soil fields (Kleczek et al., 2017). For example, Bonekamp et al., (2018) found that precipitation is extremely sensitive to the spinup time in summer, with the best performance coming with 24 hours of spinup, while does not show a clear trend with increasing spinup time over 24 hours. For our study, it is not known *a priori* how the model simulation will be effected by the spinup time. So we conduct experiments with different spinup times

ranging from 1 day to 26 days, and investigate the influence of spinup time on the amount of precipitation, soil moisture and outlet discharge of the extreme event in the study. An overview of the initialization experiments performed in the paper is given in Table 2. The evaluation period is 23 - 31 October 2014, and includes a minor peak flow on the 24[th] of October before the major peak flows on 26[th] and 28[th].





### 3.6 Prescribed snow cover experiments

In the October 2014 flood event, temperatures in the mountains were above freezing and the ground was bare. In other words, there was no layer of snow to act as a sponge and potentially affect the discharge. In a future warmer climate, however, rain-on-snow events are likely to increase, especially in mountainous areas of Norway (Vormoor et al., 2016). However, the potential impact of snow conditions on extreme flows is not well known. Therefore, we construct a series of hypothetical experiments for a primary check on this impact. The results can be helpful for filling this knowledge gap and
dictating the flood generation processes for Norway, although we know this hypothetical case most likely will increase in eastern and northern Norway instead of western Norway.

In the study, we perform two types of snow experiments: (1) different uniform snow depths are applied over the entire study area (i.e., 0.1m, 0.5m, 1m and 2m) and (2) 1m of snow is imposed above certain elevations (i.e. 400m, 600m and 800m above sea level (a.s.l.)). The experiments are all performed with the calibrated parameter set. More details can be found in
Table 3. We are mainly interested in evaluating the precipitation-snowmelt timing and snowmelt augmentation of the peak flow, if any. Therefore, we apply the prescribed snow cover fields in the restart file on the 25$^{th}$ of October 2014, which is from the 26-day spinup experiment. The area-elevation distribution in the four selected catchments is shown in Table 4. The Kinne and Myrkdalsvatn catchments are dominated by higher elevations with 48% and 44% of the area above 1000m a.s.l., respectively, compared to 36% and 9% from Bulken and Svartavatn (Table 4).

## 4 Results

### 4.1 WRF-Hydro discharge calibration

Since calibration is computationally demanding we calibrate WRF-Hydro based on the discharge of Svartavatn, which is the smallest catchment in the study region. The remaining three catchments are used for model validation. The calibration and validation results are shown in Table 5. The Nash-Sutcliffe-Efficiency coefficient (NSE) of daily discharge increases from
0.31 to 0.86, while the Bias and RMSE decrease from 6.74 to 0.56 and from 1.2 to 0.55, respectively. It indicates that the calibration greatly improves the representation of discharge over Svartavatn. The NSE values are 0.77, 0.80 and 0.76 from Bulken, Kinne and Myrkdalsvatn, which are satisfactory although they are slightly lower than the NSE of 0.86 from Svartavatn. The infiltration parameters (refdk and refkdt) and Manning routing coefficients (mn01) are calibrated by PEST auto-calibration approach to be 3.82E-6, 0.63 and 0.18, respectively. The correlation coefficient values of mn01 and refdk,
mn01 and refkdt, refdk and refkdt are -0.23, -0.16 and 0.90, respectively. We can see that there is a high correlation between two infiltration parameters of refdk and refkdt. Figure 3 shows the daily observed discharge (black line) and simulated WRF-Hydro discharge from four study basins using various refkdt values for the extreme event during 23 - 31 October, 2014. It can be seen that WRF-Hydro is sensitive to the infiltration parameter refkdt and the uncertainty of peak flow is related to the



parameters' uncertainties. The peak discharge decreases from 717, 309, 83 and 102 m³/s with REFKDT of 0.2 to 698, 217,
81 and 85 m3/s with REFKDT of 2.0 at Bulken, Kinne, Svartavatn and Myrkdalsvatn basin, respectively. An increase of
refkdt in WRF-Hydro modelling leads to a decrease in peak flow, while a slower recession limb in the hydrograph.

The daily observed discharge and simulated discharge based on the calibrated parameter set and the non-calibrated parameter
set from the four study catchments are plotted in Figure 4. The hydrographs show that the calibrated runs capture the peak
timing and magnitude well in all four catchments and that calibration markedly improves these features. The water balance
of the four study catchments is shown in Table 6, which shows that the discharge at the four study catchments is driven by
intense rainfall, and the impact of evapotranspiration and the changes in snow depth water equivalent and soil moisture are
minor.

**4.2 Benchmark evaluation**

Furthermore, the benchmark model efficiencies are also shown in Table 5. Daily precipitation, temperature and potential
evaopotranspiration from WRF are used as input to the HBV light model in order to calculate benchmarks. For the upper
benchmark (i.e. using calibrated parameters), the calibrated HBV model efficiency ($R_{upper}$) of Svartavatn basin is 0.80. For
the lower benchmarks, the HBV model efficiency is 0.43 when calculated from random parameters ($R_{lower/random}$) and is 0.67
when calculated from regionalized parameter sets based on three nearby catchments ($R_{lower/regional}$). Regarding the Bias and
RMSE values, they are -0.42 and 9.03 from calibrated WRF-Hydro, 2.52 and 11.3 from the upper benchmark, and 7.56 /
2.95 and 18.43 / 14.13 from two lower benchmarks ($R_{lower/random}$ / $R_{lower/regional}$). These results show that the calibrated WRF-
Hydro model NSE of 0.86 is well above the upper benchmark (0.80). Besides, the calibrated WRF-Hydro has both less bias
and smaller RMSE than the upper benchmark. Despite this encouraging result some care must be taken in the interpretation
due to uncertainty in the input data for the HBV simulation caused by a lack of long-term averaged monthly meteorological
forcing.

**4.3 Precipitation evaluation**

The accumulated precipitation from 23 October at 06 UTC to 31 October at 06 UTC is shown in Figure 5. The CTRL
simulation (see Table 3) is shown in colored contours and the observed values in colored squares (circles) for the HOBO rain
gauge (meteorological) observational network. The observed precipitation amounts correspond well with the model
simulation at the majority of the stations. The spatial variability is large in the complex terrain with several areas receiving
close to 500mm precipitation and some areas less than 100mm during the week.

The temporal evolution of simulated precipitation at monitoring locations is shown in Figure 6. Observational stations are
depicted in Figure 6a and the simulated precipitation interpolated from the four nearest grid points in the CTRL simulation is





shown in Figure 6b. Daily precipitation values are shown as diamonds, whereas hourly values are shown as continuous lines. The temporal evolution is generally well reproduced by the simulation, as well as the timing of the precipitating periods.

**4.4 Sensitivity to spinup time**

Five different spinup times are investigated in order to analyze the sensitivity of precipitation and discharge to the initial conditions (see list in Table 3). The same calibrated parameter set is used for all the spinup experiments.

During this event the western coast of Norway was exposed to a considerable amount of precipitation within a 4-day period. Further, the soil was already saturated after a wet October. This can be seen in Figure 7 which shows (a) the catchment

averaged total soil moisture of the four catchments in the CTRL experiment (26d-spinup), (b) the averaged total soil moisture on 24 October, (c) the difference of soil moisture between the 1d-spinup and CTRL, and (d) the difference of soil moisture between the 12d-spinup and CTRL. Figure 7 indicates a sensitivity of soil moisture to spinup time although the differences are fairly small (-10 to 10 mm). In general, the soil becomes slightly more wet with increased spinup time.

In addition, we evaluate the temporal evolution of the precipitation for the spinup experiments. Figure 8 shows the

accumulated precipitation interpolated from the four nearest grid points to the following four rain gauge stations, Øvstedal, Myrkdalen, Mjølfjell and Vossevangen. These are the official meteorological observational stations located in the catchments of Svartavatn, Myrkdalen, Kinne and Bulken, respectively. The precipitation sensitivity to spinup time is low in all catchments. At Øvstedal and Mjølfjell the precipitation is reproduced well, whereas the remaining two stations are somewhat biased. The model seems to be unable to catch the finer scale phenomena completely with a 3 km grid resolution

especially over a small complex catchment such as Myrkdalvatn. This could be partly due to the combination of highly complex terrain and interactions with the Sogne Fjord, just to the north, that are missing in the simulation. Table 7 provides additional evidence of this, showing the mean absolute error (MAE) in the total accumulated precipitation and discharge. The differences in MAE of precipitation between the spinup experiments are negligible; we believe this is because of the large-scale nudging in the outer domain. Besides, there is no decrease in precipitation MAE with the spinup time increase in

any of the stations, except Øvstedal. The averaged MAE of precipitation at all 54 observational stations in the area is around 50 mm. Especially in Kinne, the discharge MAE increases with the extending of the spinup time. This suggests that the model, even at 3 km grid spacing, struggles to fully reproduce the local-scale orographic effects in the complex terrain around Voss and Myrkdalen. A previous study in the high mountains of Asia, suggested that a sub-kilometer grid is needed for accurately estimating truly local meteorological variability (Bonekamp et al., 2018).

The temporal evolution of streamflow over the four catchments is shown in Figure 9, which shows the daily hydrograph of discharge for the four catchments with different spinup experiments. We want to keep this flood event completely in our spinup experiment, so we evaluate the period of 23 - 31 October. From the results, we can see that the precipitation amount and timing do not differ significantly between spinup times in any of the catchments. However, the discharge at the pre-flood



phase, which is 23–24 October, is more sensitive to the spinup time. For Svartavatn this sensitive phase even extends to the

26<sup>th</sup>. This is because the initial condition of soil moisture affects the overland flow that dominates the discharge of this catchment. The pre-flood discharge moves closer to the observed discharge when we increase the spinup time from 1 day to 26 days. In general, the peak flows are overestimated compared with the observations, except in the Svartavatn catchment. This is because we only calibrated the model in Svartavatn, and then used this calibrated parameter set in the simulation for the other three catchments, which perforce have poorer performance than the Svartavatn catchment.

**4.5 Sensitivity to prescribed snow cover**

The dynamical modelling experiments of different snow conditions are based on the WRF-Hydro simulation with a 26-day spinup, which is labelled as the control (CTRL) in the snow experiments (Table 2). In these experiments, the snow depth and the water equivalent snow depth are changed in the 25th of October restart file and the simulations are restarted in offline mode. An overview of the snow experiments is presented in Table 3. To summarize: one set of experiments tests the

sensitivity to varying snow depths while the other set of experiments tests the sensitivity to snow elevation. The temporal evolution of catchment averaged SWE is shown in Figure 10 and Figure 11. A decline throughout the simulation period is shown for all catchments. Unsurprisingly, the snowmelt is due to positive surface temperatures and precipitation. The experiments with the 0.5m, 1m and 2m snow depths have similar snowmelt behavior during 25 - 28 October. The snowmelt stops in all the snow experiments after 29 October, because of a drop in both rainfall and temperature (below 273K), which

can be seen in Figure 10. More detailed information of the total snowmelt during 25 - 31 October under the different snow experiments is given in Table 8. From the table, we can see that, except for the 0.1m snow depth experiment where the added snow quickly melts away, the results from other three prescribed snow depth experiments (0.5, 1.0 and 2.0 m) are fairly similar, where the total water equivalent snow melt are 14 -16 cm in Svartavatn, 11-12 cm in Myrkdalsvatn, 11-12 cm in Kinne and 12-13 cm in Bulken. This is because the limit of melting snow is controlled by the temperature and precipitation

and maximum of around 0.5 m snow will be melted away in this case. For the snow elevation experiment where 1 m of snow were added above the given ground elevation, the response is a result of the elevation of the catchments. In Kinne and Myrkdalsvatn there is little variation, around 8-11 cm of SWE is melting. Their average catchment height is so high that there is only a small difference in total SWE between the experiments, leading to similar response. For Svartvatn and Bulken the situation is different. The total SWE in the catchments vary between the experiments because of the lower catchment

elevation, hence the resultant SWE melting varies between 2-16 cm for Svartvatn and 4-12 cm for Bulken.

Figure 12 shows the hydrograph of hourly discharge for the experiments where snow depth is modified uniformly over the entire area. From both Figures 9 and 12, we can see that the difference in melt between the shallowest and deepest snow depths is less than 2 cm, which suggests that snow depths beyond 0.5m did not contribute markedly to more discharge. The main contribution from the additional snow is to enhance the peak discharge in all the catchments. Also, the contribution





from melting snow is mostly confined to precipitating periods, which also coincides with higher temperatures. The fact that snow depths above 0.5m have little impact suggests that rain-on-snow can melt at most 0.5 m snow under this experiment design. The snowmelt discharge decreases after 29 October, which is preceded by a drop in both rainfall and surface temperature (below 273k). It is worthwhile to recall that the Noah LSM has a simple bulk snow-soil-canopy layer model. Previous studies have noted that there was a positive bias in snow surface energy in the Noah LSM, which resulted in an

underestimation of snow water equivalent (SWE), and led to a reduced snow pack during winter and earlier snowmelt in spring (Jin and Miller, 2007; Jin and Wen, 2012; Niu et al., 2011). In our experiments this positive bias in snow surface energy in the Noah LSM together with energy from intense rainfall are probably used to melt snow directly, so that the warm snowpack doesn't retain any liquid water which can refreeze during the day before it runs away. In this case, we might see unrealistically high snowmelt and the snowpack would not act like a sponge and retain part of the rainfall. All in all, the

snow experiments show that intense precipitation coinciding with higher temperature can result in up to 0.5 m of snow melt, which contributes to the peak flow. However, more work needs to be done in the future with more sophisticated multi-layer snow models to confirm the behaviour observed in these idealized experiments.

The effects of varying snow cover by altitude on daily discharge are shown in Figure 13. Here, we perform snow experiments where 1m of snow is imposed above certain elevations, i.e., 400, 600, and 800 m. Those prescribed snow covers

are applied in the restart file on 25 October 2014, which is from the 26-day spinup experiment with calibrated parameter set. From Figure 13 we can see that (a) there are increases in snow-melt runoff with the elevation decreases from 800 m to 0 m; (b) the differences of snow-melt runoff among different experiments vary in different catchments, for example, the snow-melt runoff from 0 m and 800 m experiments show a large difference in catchment Svartavatn, while not much difference can be seen in catchment Kinne and Myrkdalsvatn. This is because varying the prescribed snow cover by elevation has a

greater influence on the lower catchments, i.e., Svartavatn (with 61% of the area below 800 m), than the higher catchments such as Kinne and Myrkdalsvatn (with 69% and 71 % above 800 m, respectively). A more detailed quantitative estimate of the total water equivalent snow depth change during 25-31 October under different prescribed snow cover experiments is given in Table 8. It confirms the results in Figure 10 and Figure 11, with the first half-meter of the snowpack contributing the to the snowmelt. In addition, there is a greater SWE decrease in the lower elevation catchment Svartevatn (i.e., -0.16 m in

the added 1 m snow experiment) than Kinne and Myrkdalsvatn catchments (i.e., -0.11 m in the added 1 m snow experiment), which are dominated by higher elevations.

## 5    Discussion and conclusions

In this study, we aimed to reproduce an extreme weather event in a region characterized by complex terrain. A dynamical hydrometeorological modeling system (WRF-Hydro) was employed for this purpose. A nested WRF atmospheric model, run





at convection permitting scales, was used to reproduce the meteorological event and provide precipitation forcing for a distributed hydrological model over a small domain encompassing four study catchments affected by extreme flooding. 3km grid spacing was used for the WRF atmosphere and land surface while a 300m grid spacing was used for the WRF-Hydro river routing. An auto-calibration tool was used for WRF-Hydro model calibration based on the daily discharge at Svartevatn, which is the smallest of the 4 catchments. The simulation of high-resolution precipitation and discharge were

assessed based on observational data sets. Also, the sensitivity of the results to the spinup time and snow depth was investigated.

The results show that the precipitation from the 3km simulation generally agrees well with the rain gauges both in terms of temporal evolution and spatial variability, although it underestimates the precipitation in the highly complex terrain around Myrkdalen. This underestimation could be due to a combination of the locally complex topography and the proximity to the

Sognefjord only 15 km away. This large, but narrow body of water, and its many offshoots, is not well-resolved by the modeling system.

The auto-calibration greatly improves the model performance with the NSE increasing from 0.41 to 0.86, bias and RMSE decreasing from 5.29 and 19.05 to -0.42 and 9.03, respectively. The modeling system captures peak flow volumes and timing well after model calibration. Besides, WRF-Hydro runoff performance is depends on some highly sensitive parameters (e.g.

infiltration parameter and Manning routing coefficients).

Comparing with the benchmarks, the calibrated WRF-Hydro NSE value (0.86) is higher than the upper benchmark from the HBV light model (0.80). This might be due a lack of long-term input data (e.g., averaged monthly potential evapotranspiration). The implication is that WRF-Hydro may perform as well or even better than a simpler conceptual hydrological model, especially for ungauged basins or observation scarce regions.

The precipitation simulation is not overly sensitive to spinup time. We find that mean absolute errors of precipitation are very similar given different spinup times. This could be due, in part, to the decision to nudge the atmospheric flow to match that large-scale reanalysis. Discharge simulations are slightly more sensitive to the spinup time due to the impact of soil moisture, especially during the pre-peak phase. We find a spinup time of 26 days give the lowest MAE of precipitation and discharge compared to the other smaller periods.

SWE melt during 25-31 October is consistently around 10 ~16cm for the uniform snow depth experiments (0.5 - 2 m). The results also show that melting snow contributes most to discharge during the rainy periods and the peak flow periods. This indicates that snow cover intensified the extreme discharge instead of acting as sponge in this study, which suggests that future rain-on-snow events may potentially result in higher flood risk. However, more sophisticated snow models and targeted experiments should be conducted to confirm this speculation.

Our results increase confidence in the performance of WRF-Hydro for simulating extreme hydrometeorological events over complex terrain. Further, they demonstrate the importance of model calibration and reasonably accurate land surface initial





conditions for simulating discharge, especially for peak flow. The snow experiments suggest that rain-on-snow events under warmer conditions may contribute to an increase in flood magnitudes in Norway, due to projected increases in extreme precipitation (Lawrence, 2016). However, targeted experiments on the changing risks associated with future rain-on-snow

events are needed to confirm this possibility.

**Acknowledgments**

This study was joint funded by the Research Council of Norway (RCN) project EVOGLAC (grant 255049), HordaKlim (grant 245403) and R3 (grant 255397), by the Bjerknes Center for Climate Research (SKD) project WACYEX and CHEX. The computer resources where available through the RCN's program for supercomputing (NOTUR/NORSTORE); projects

NN9280K, NN9478K and NS9001K. All WRF-Hydro simulation data in this paper are available from the authors upon request (luli@norceresearch.no ).

*Author contributions.* LL contributed to most of the modelling, analysis, writing and revising of the paper. MT contributed to collecting HOBO data, analysis and assisted with writing and reviewing the paper. SS contributed to the reviewing the paper.

AS contributed to the model calibration by PEST and reviewing the paper.

*Competing interests.* The authors declare that they have no conflict of interest.

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





**Table 1: The list of observed discharge stations**

| Station | Area (km²) | Latitude | Longitude |
|---|---|---|---|
| Bulken(Vangsvatnet) | 1092.04 | 6.29 | 60.63 |
| Kinne | 511.8 | 6.50 | 60.63 |
| Svartavatn | 72.3 | 5.90 | 60.65 |
| Myrkdalsvatn | 158.87 | 6.50 | 60.80 |

**Table 2: Overview of performed initialization experiments. All experiments are based on the calibrated parameter-set.**

| Name | Spinup time |
|---|---|
| 1d-spinup | 1 day |
| 3d-spinup | 3 days |
| 5d-spinup | 5 days |
| 14d-spinup | 14 days |
| 26d-spinup* | 26 days |

*: 26d-spinup is used as the control (CTRL) experiment in the snow experiments.


**Table 3: Overview of pre-existing snow cover experiments. All experiments are performed during 25 - 31 October, based on the 26 days spinup simulation with calibrated parameter-set.**

| Name | Added snow depth (m) | Added water equivalent snow depth* (m) | Elevation limits for adding snow** (a.s.l. m) |
|---|---|---|---|
| 2m snow | 2 | 0.67 | 0 |
| 1m snow | 1 | 0.33 | 0 |
| 0.5m snow | 0.5 | 0.17 | 0 |
| 0.1m snow | 0.1 | 0.03 | 0 |
| 800m Elev | 1 | 0.33 | 800 |
| 600m Elev | 1 | 0.33 | 600 |
| 400m Elev | 1 | 0.33 | 400 |
| 0m Elev | 1 | 0.33 | 0 |

\* the snow density is assumed to be 300 kg/m^3
\*\* the elevation limits for adding snow is 0 m, which means the snow is added over the whole catchment area.






**Table 4: Area-elevation of the four catchments**

| >Elevation | 400 m | 600 m | 800 m | 1000 m |
|---|---|---|---|---|
| Bulken (%) | 85 | 73 | 57 | 36 |
| Kinne (%) | 92 | 85 | 69 | 48 |
| Svartavatn (%) | 89 | 67 | 39 | 9 |
| Myrkdalsvatn (%) | 94 | 83 | 71 | 44 |

Table 5: WRF-Hydro uncalibrated, calibrated, validated results with model efficiency values for different benchmarks

| | Catchment | NSE | Bias(mm) | RMSE(mm) |
|---|---|---|---|---|
| **Default (non-calibration)** | Svartavatn | 0.41 | 5.29 | 19.05 |
| **Calibration** | Svartavatn | 0.86 | -0.42 | 9.03 |
| **Upper benchmark** | Svartavatn | 0.80 | 2.52 | 11.30 |
| **Lower benchmark (random parameter values)** | Svartavatn | 0.43 | 7.65 | 18.43 |
| **Lower benckmark (regional parameter values)** | Svartavatn | 0.67 | 2.95 | 14.13 |
| **Validation** | Bulken | 0.77 | 0.99 | 6.64 |
| | Kinne | 0.80 | -1.52 | 5.80 |
| | Myrkdalsvatn | 0.76 | -0.005 | 6.65 |

**Table 6: The water balance of four study catchments during 25 - 31 October 2014 (based on calibrated parameter sets, unit: mm)**

| Catchments | Observed Discharge | Precipitation | Discharge | ET | Soil water | Water equivalent snow depth | Residuals |
|---|---|---|---|---|---|---|---|





| | | | | | | | |
|---|---|---|---|---|---|---|---|
| Svartavatn | 315 | 293.6 | 305.7 | 1.5 | 3.2 | 0.3 | -17.1 |
| Bulken | 243.2 | 230.3 | 236.7 | 2.1 | -2.5 | 0.9 | -7 |
| Kinne | 187.6 | 206.6 | 219.8 | 1.2 | -11.7 | 1.2 | -3.9 |
| Myrkdalsvatn | 215 | 211.1 | 208 | 1.9 | -14.7 | 0.6 | 15.3 |


**Table 7: Mean absolute error [mm] of precipitation and discharge compared with observations under impact of spinup time (23-31 October). The accumulated precipitation was interpolated from the four nearest grid points to Øvstedal, Myrkdalen, Mjølfjell, Vossevangen and (ALL) the 54 observational stations available in the area.**

| MAE [mm] | spinup | Øvstedal (Svartavatn) | Myrkdalen (Myrkdalsvatn) | Mjølfjell (Kinne) | Vossevangen (Bulken) | ALL |
|---|---|---|---|---|---|---|
| | 1d | 20 | 115 | 31 | 72 | 51 |
| | 3d | 19 | 114 | 36 | 71 | 50 |
| **Precipitation** | 5d | 5 | 113 | 34 | 69 | 52 |
| | 12d | 5 | 114 | 31 | 73 | 51 |
| | 26d | 5 | 125 | 32 | 69 | 51 |
| | 1d | 108 | 35 | 9 | 53 | 29 |
| | 3d | 80 | 32 | 13 | 48 | 20 |
| **Discharge (m3/s)** | 5d | 56 | 19 | 29 | 35 | 11 |
| | 12d | 59 | 17 | 26 | 31 | 13 |
| | 26d | 37 | 19 | 23 | 35 | 8 |


**Table 8: The total water equivalent snow depth change during 25-31 October under different pre-existing snow cover experiment in four study catchments.**





| Experiments | Svartavatn | Myrkdalsvatn | Kinne | Bulken |
|---|---|---|---|---|
| CTRL | 0.0004 | 0.0006 | 0.0014 | 0.0012 |
| 2m snow | -0.16 | -0.12 | -0.12 | -0.13 |
| 1m snow* | -0.16 | -0.11 | -0.11 | -0.12 |
| 0.5m snow | -0.14 | -0.11 | -0.11 | -0.11 |
| 0.1m snow | -0.02 | -0.02 | -0.02 | -0.02 |
| 800m Elev | -0.02 | -0.08 | -0.08 | -0.04 |
| 600m Elev | -0.09 | -0.09 | -0.10 | -0.09 |
| 400m Elev | -0.12 | -0.11 | -0.11 | -0.11 |
| 0m Elev* | -0.16 | -0.11 | -0.11 | -0.12 |

*: 1m snow and 0m Elev are the same experiment.




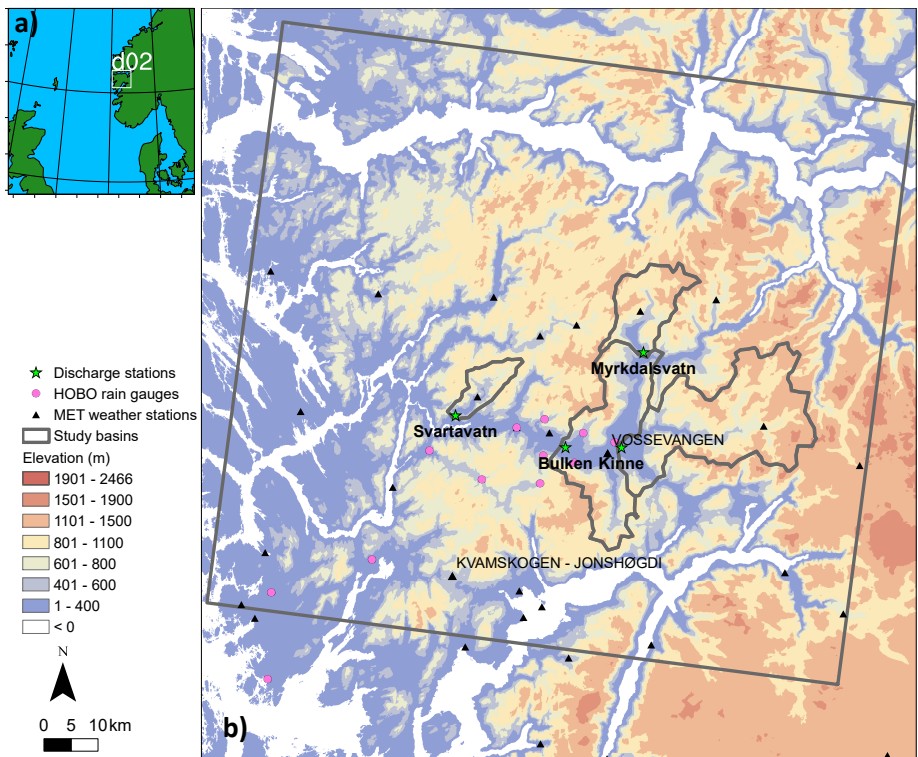

**Figure 1: The study catchments locate at western Norway. (a) Outer domain (resolution 9 km), borders of the inner domain (d02)**
**are shown with a white frame; (b) zoom on the inner domain (resolution 3 km); topography and four study catchments boundaries are shown with details about ground-based observation stations, including discharge stations, HOBO rain gauges and MET weather stations (from met.no), for model calibration and further performance evaluation.**





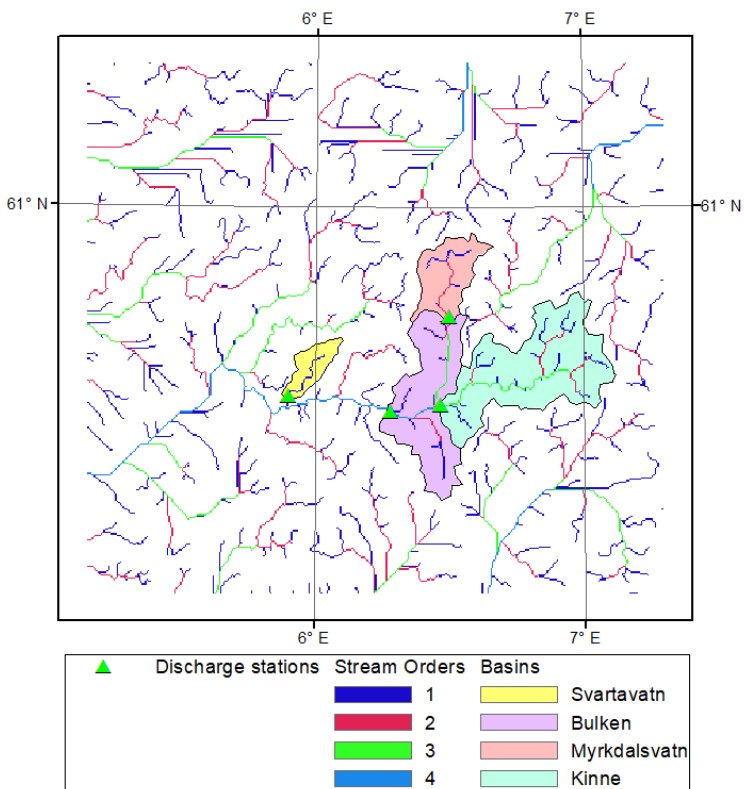

**Figure 2: Map of physiographic grid for routing processes with the four study catchments (i.e., Svartavatn, Bulken, Myrkdalsvatn and Kinne), the discharge stations and stream orders in the inner domain from hydrological modelling in WRF-Hydro model system. It is worth to note that the Bulken is at the downstream of Kinne and Myrkdalsvatn, which means the drainage basin of Bulken catchment comprises the catchment of Kinne (pink) and Myrkdalsvatn (light green) in the study.**

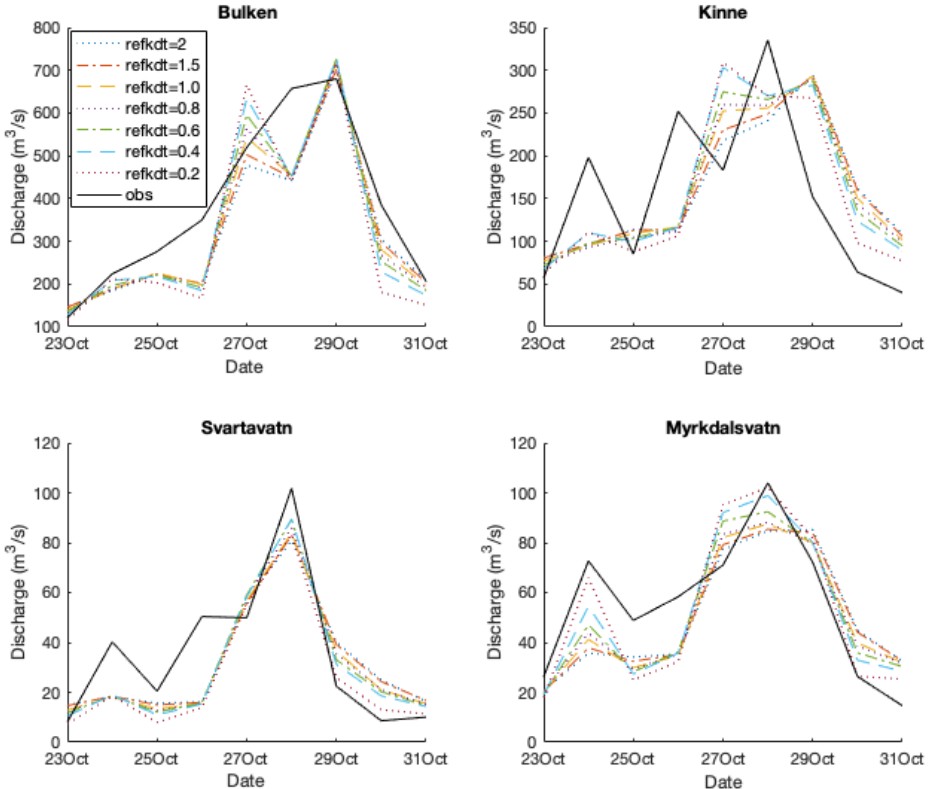

**Figure 3: Hydrographs of daily observed discharge (obs) and simulated WRF-Hydro discharge from four study basins using various refkdt values for the extreme event during 23 - 31 October, 2014.**





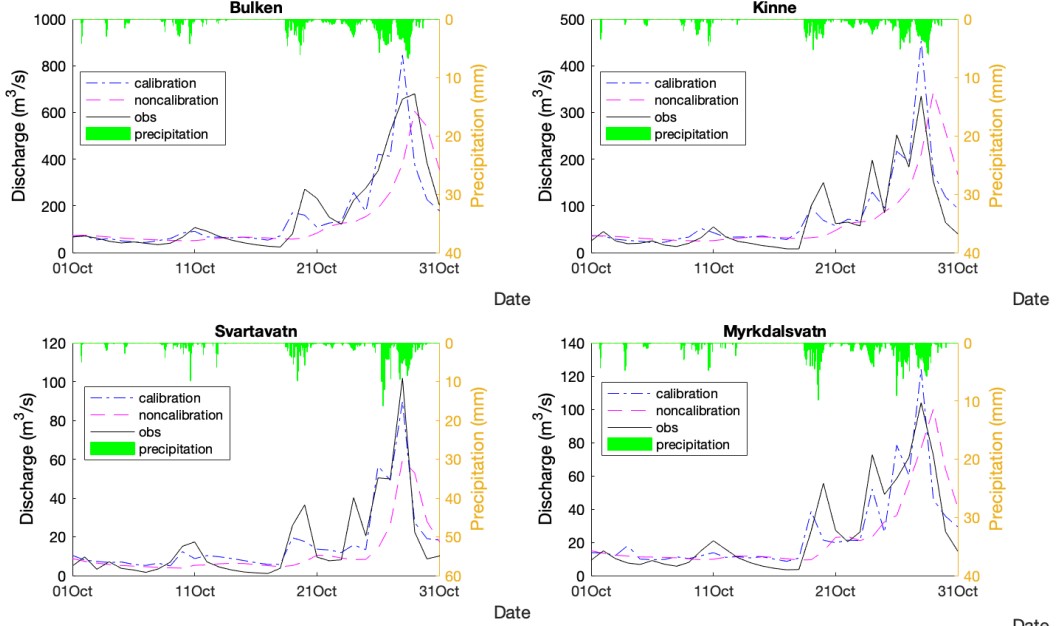

**Figure 4: Hydrographs of daily observed discharge (obs) and simulated discharge with calibrated parameters (calibration) and non-calibrated parameters (noncalibration) with precipitation during 1 - 31 October, 2014.**

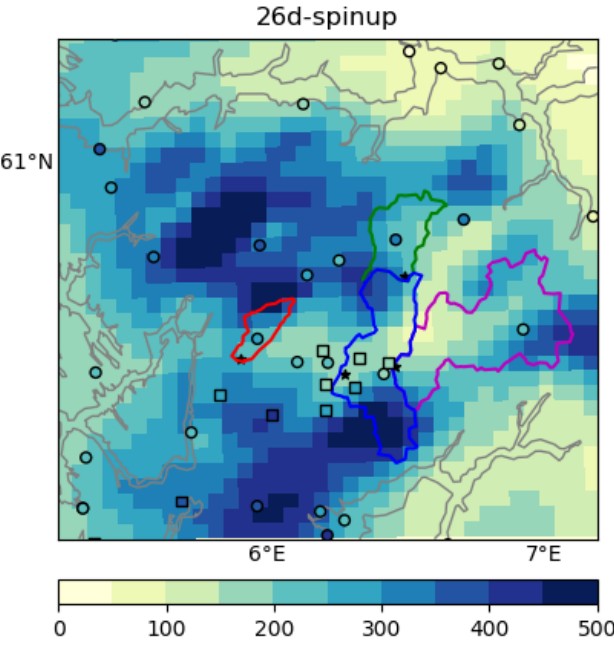

**Figure 5: Accumulated precipitation (mm) during 23 - 31 October from the 26 day spin up CTRL simulation. The catchments are contoured in colours corresponding to Figure 8. Squares and circles are observational values from the hobo network and the meteorological network respectively.**



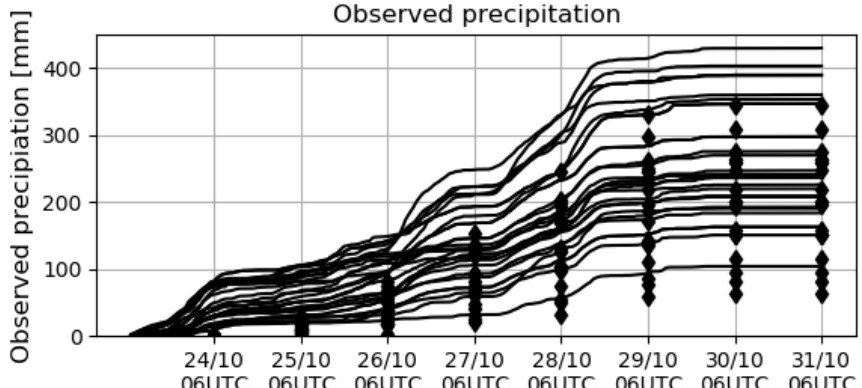

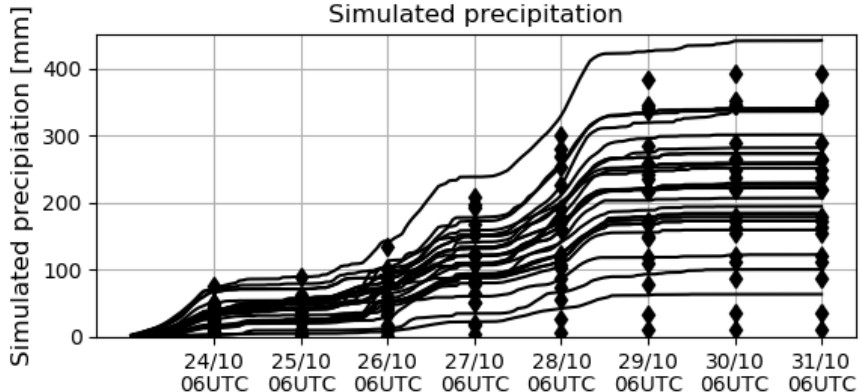

**Figure 6: Accumulated precipitation from 23 October at 6 UTC to 31 October at 6 UTC a) at observation stations in the area and b) interpolated from the four nearest grid points to the equivalent station position in the 26d spin up simulation. The lines represent stations with hourly precipitation measurements whereas the diamonds represent stations with daily precipitation values. The same notation is used in b), though the model output enables higher temporal resolution at the daily station positions.**


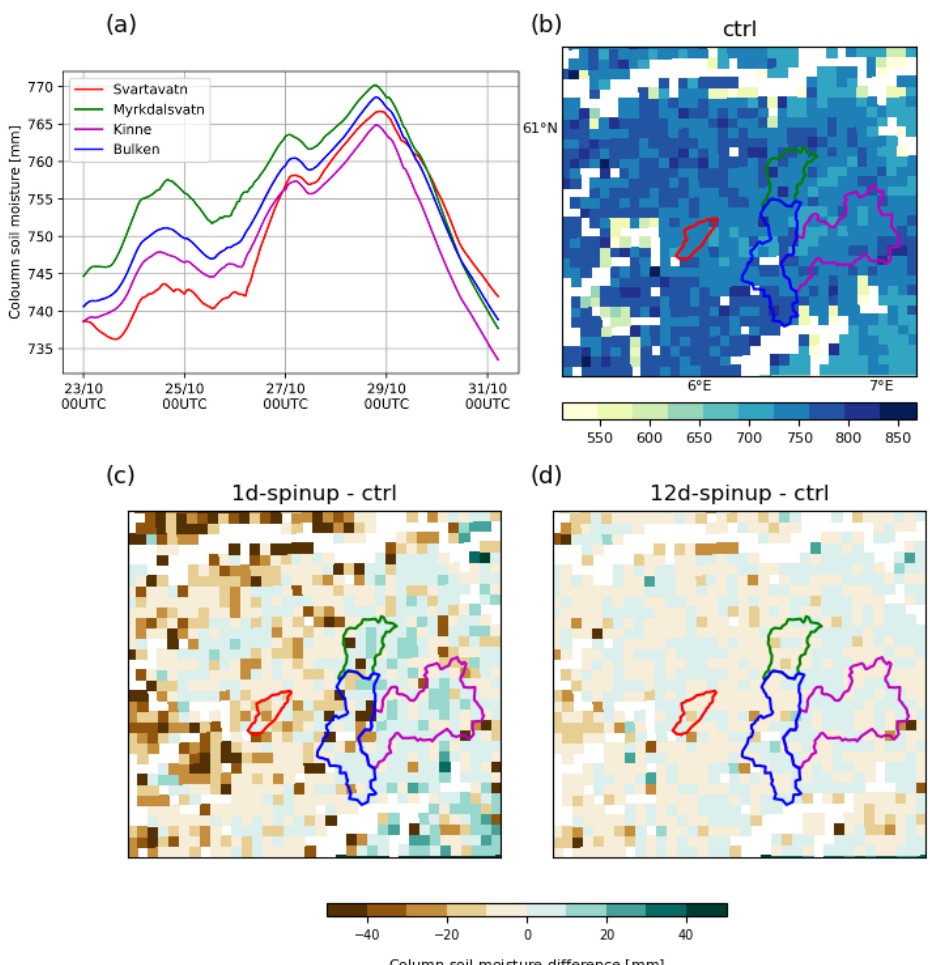

**Figure 7: a) Temporal evolution of total column soil moisture averaged in the catchments (ctrl simulation); b) Averaged total column soil moisture on 24 October; c) soil moisture differences between the 1-day spinup and the control simulation; d) soil moisture difference between the 12-day spinup and the control simulation.**





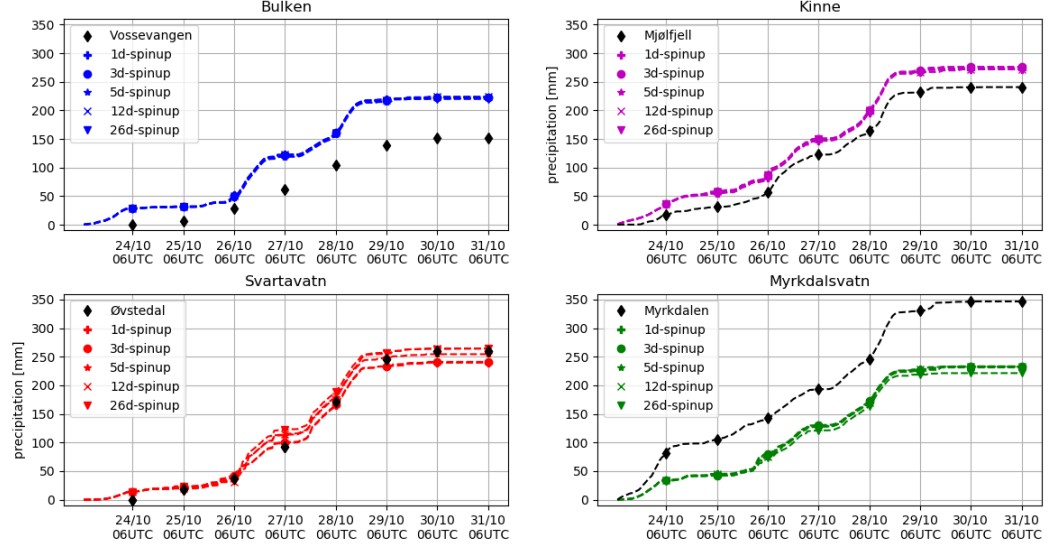


**Figure 8: The accumulated precipitation from the simulations at the four nearest grid points to Vossevangen (Bulken), Mjølfjell (Kinne), Myrkdalen (Myrkdalsvatn) and Øvstedal (Svartavatn), observational stations from the simulations of different spinup times, including 1-day (1d-spinup), 3-day (3d-spinup), 5-day (5d-spinup), 12-day (12d-spinup) and 26-days (26d-spinup) compared with the observational station within the catchment.**





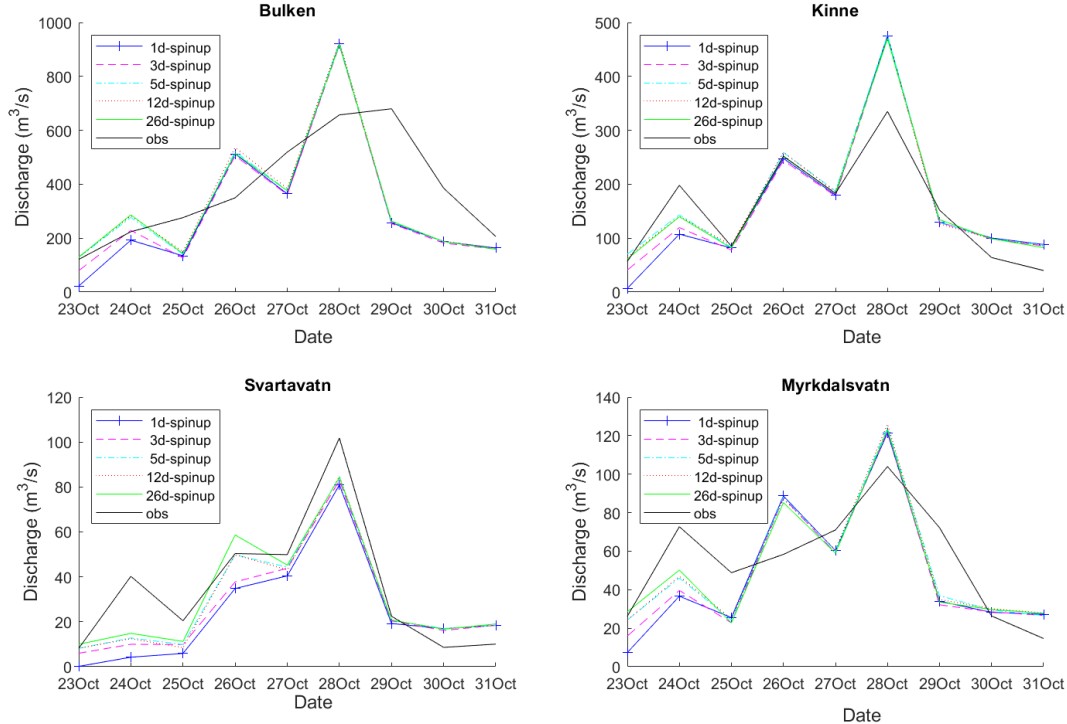


**Figure 9: Daily observed streamflow (obs) and simulations discharges with different spinup time, including 1 day (1d-spinup), 3 days (3d-spinup), 5 days (5d-spinup), 12 days (12d-spinup) and 26 days (26d-spinup) for the flooding events during 23 - 31 October, 2014.**





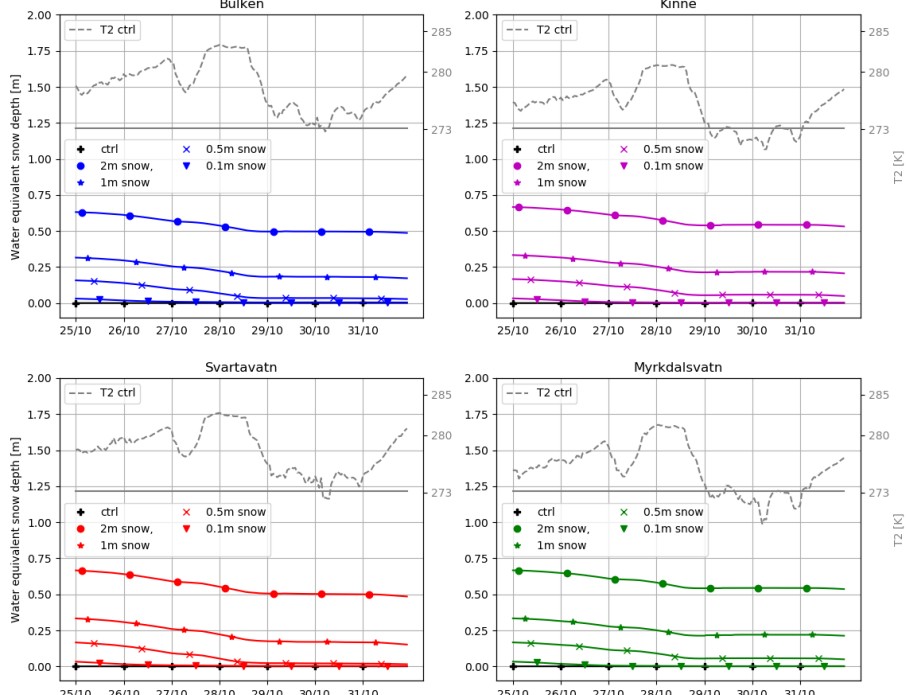

**Figure 10: Catchment averaged hourly snow water equivalents from 3km WRF-Hydro simulation for the snow sensitivity experiments, including 2 m added snow depth (2m snow), 1 m added snow depth (1m snow), 0.5 m added snow depth (0.5m snow), 0.1 m added snow depth (0.1m snow), and the catchment averaged hourly snow water equivalent (ctrl) and surface temperature (T2) from the control simulation of the 26 day spinup.**




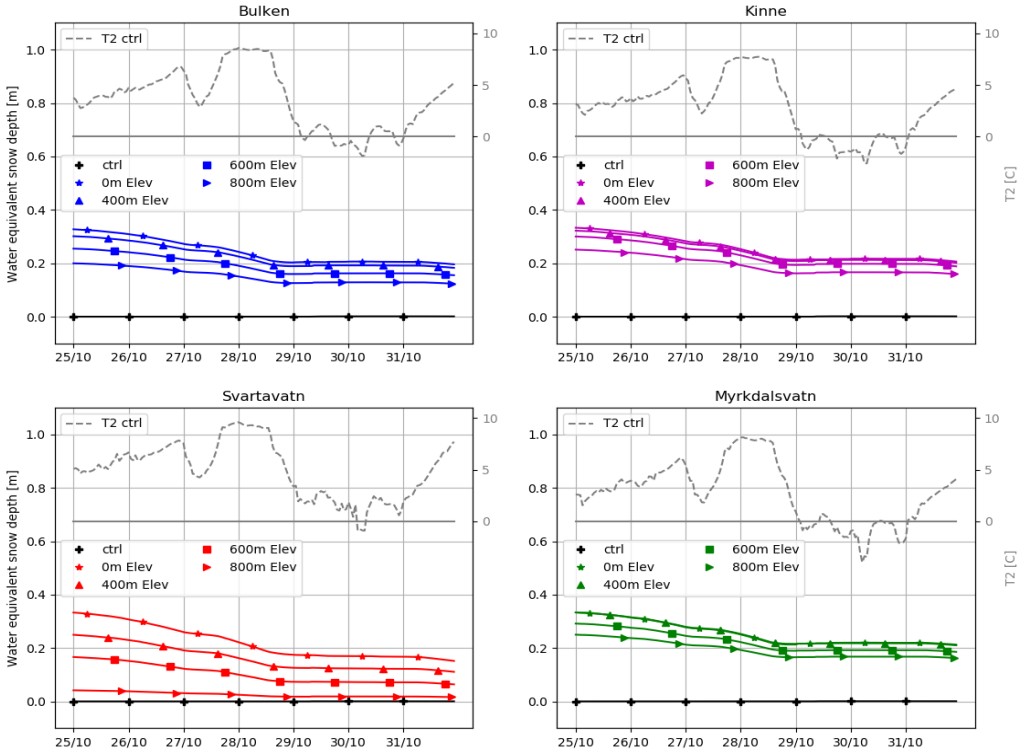

**Figure 11: Catchment averaged snow water equivalents from 3km WRF-Hydro simulation for the pre-existing snow cover sensitivity experiments, including 1 m added snow depth to the area where the elevation is above 0 m (0m Elev), 400 m (400m Elev), 600 m (600m Elev), and 800 m (800m Elev) based on the CTRL simulation (26 day spinup) ), and the catchment averaged hourly snow water equivalent (ctrl) and surface temperature (T2) from the control simulation of the 26 day spinup.**





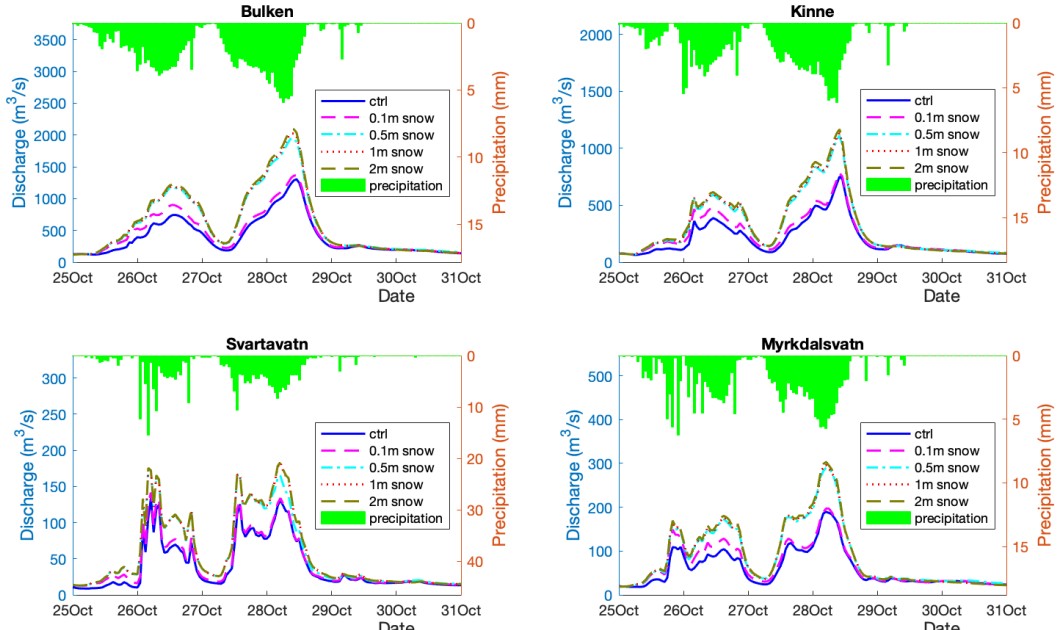

**Figure 12: Hourly simulated discharge during 25 - 31 October from different experiments of added snow depths, i.e. 0.5 m, 1 m and 2 m snow and the control simulation from 26-day spinup without added snow depth (ctrl).**



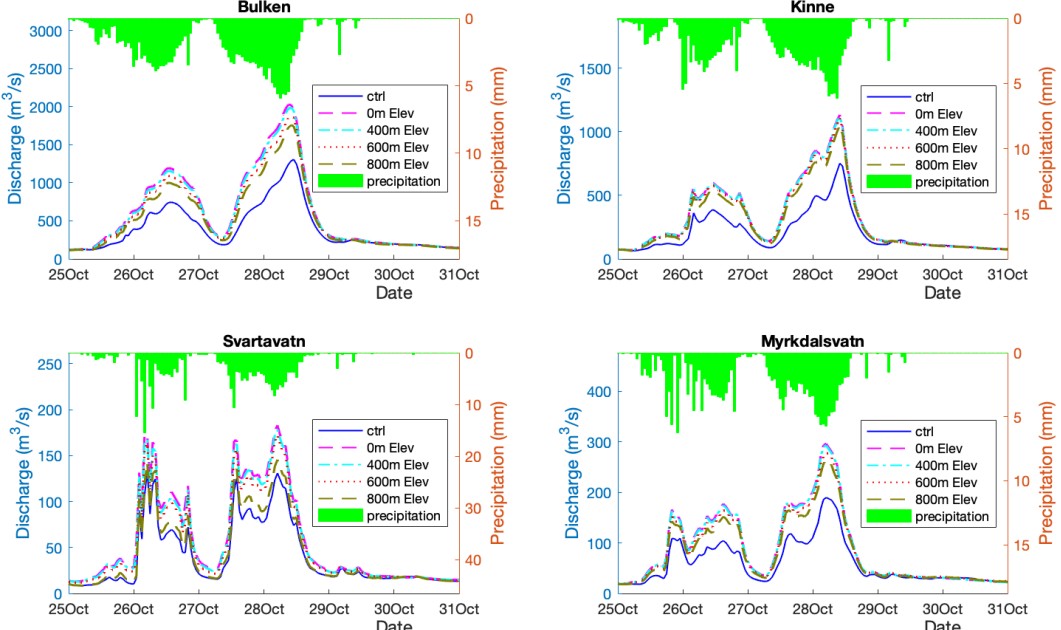

**Figure 13: Hourly simulated discharge during 25 - 31 October from the control simulation (ctrl) and the different experiments of added 1m snow depths on the areas with elevation above 0, 400 m, 600 m and 800 m.**