# Peer review of "The impact of initial conditions on convection-permitting simulations of a flood event over complex mountainous terrain"

_Hydrology and Earth System Sciences, 2019_

## Referee Comment (RC1) · Anonymous Referee #1 · 6 Nov 2019

Review of manuscript "The impact of initial conditions on convection-permitting simulations of flood events " (hess-2019-402) by Li et al.

The manuscript is well written and clearly shows the impact of spinup time and different snow cover on discharge modeling with WRF-Hydro. This work is of high interest for the hydrological modeling community. It should therefore be considered for publication in nhess.

Minor comments For clarity and conciseness I think the size of the abstract should be reduced to less than 300 words (currently 438 words)

Page 10, line 285: add units to the values of bias and RMSE. By the way, please consider formulating the bias in percentages instead of mm, in order to facilitate the

interpretation of the results.

Page 10, line 289: I guess this "correlation coefficient" comes from the PEST method. Please clarify.

Page 11, lines 299-302: this result implies that the runoff coefficient, that is the ratio between discharge and precipitation, is slightly above 1! in order to argue that this is realistic, the authors may emphasize the fact that the case study is in "polar region" and that at the end of October not much ET is expected.

Page 12, line 333: it would be useful to also add the values in %, in order to better assess that the "differences are fairly small".

Page 12, line 346: "discharge MAE": the whole paragraph is about precipitation so I find it confusing to have this comment about discharge at this place. Is it a typo? Otherwise, this could be moved to the next paragraph.

Page 13, line 367: "Unsurprisingly, the snowmelt is due to positive surface temperatures and precipitation" In the Noah LSM, the snowmelt occurs when surface temperature is positive, independently of precipitation. Another land surface model such as NoahMP, which distinguishes between solid and liquid phase in the snowpack would be need to assess the effect of precipitation on snowmelt. So I suggest to remove the comment that the snowmelt is due to precipitation, as the Noah LSM which is used here does not allow to asses that, I think.

Page 13, line 374: same comment

Section 5: this section is rather a summary of what has already been explained in the result section, than a discussion. Please consider renaming this section, and also use the past tense.

Typos Page 3, line 87: "in to improve" –> "in order to improve"

Page 15, line 399: "1m" –> "1 m", line 406: "69%" –> "69 %", apply to the entire

manuscript

Page 15, line 429: "performance is depends " –> "performance depends"

Figures Fig. 6: labels a) and b) are missing

Fig 8: Maybe the symbols could be shifted so that they can be all seen?

[Figure]

---

## Referee Comment (RC2) · Anonymous Referee #2 · 12 Nov 2019

The comments below come from someone from the atmospheric modelling community, and I know some of the co-authors are also atmospheric model users. Hence, some comments may come from a somewhat different angle.

Overall, I find the paper quite easy to read, but there are some issues in the earlier part of the paper that the authors seem to have forgotten readership may include non-hydrologists. It will be nice to put some of the results in the broader context – like the need of high resolution models to deal with complex topography and what the snow-prescribed simulations may mean in climate change.

Major comment:

1. Be aware who might be reading the paper, and make sure they understand your

terminology. Just in the abstract and the Section 1, there are few cases that the authors use acronyms and terminologies that may be readily understandable by hydrologists but not to other readers; proper definitions somehow get mentioned later in the paper, but not on the first incidence that they are used. WRF is a tool commonly used by the NWP and climate community, so it is important to make sure all members of the WRF community to understand what you are talking about. See minor comments for details.

2. As the authors indicate that one needs the high-resolution WRF model to get the precipitation right around the complex topography in their region of interest. One way to show that is to demonstrate how different the precipitation looks like with the lower-resolution 9km model. I hope it is not too much to ask, but how would Figure 5 looks like for the 9km WRF simulation?

3. Figure 9: Is this due to the short spin up runs have soil dry enough to absorb additional precipitation into the ground during the initial phases of the event? It will be good thing to check within the context of what you see for Figure 7 and will show what the WRF land surface scheme is doing to your results.

4. If I understood Section 4.5 correctly is that the authors' sensitivity simulations with snow generally falls into two scenarios ("no or nearly-no snow" (control or 0.1 m snow) or "a lot of snow" (0.5+m). In between 0.1 and 0.5m, there will be cases that we may see snow making an impact to the discharges in between those two above bi-nary regimes. This is important in climate change impact – that the snow feedback to river flow will depend on which snow regime we will be in (no snow, a lot of snow, or somewhere-in-between regime). Of course, this is just one event and the authors have given context how rare the event is (50-year return), and it occurred in October (which would have no or little snow). Hence, it will be nice to comment on the following issues: a. Some general comments about the seasonal cycle of extremes in Norway: How often do you see comparable precipitation extreme events during late winter, spring, early summer in which snow would become a factor? Are there any reasonable reason to believe the probability of such events during the snow-relevant months to change in

the future? I hope this is just some additional literature review. b. What are the typical snow depths that you get by the end of winter? How is that expected to change? I would speculate snow depth would likely to be in "a lot of snow" regime, but it would be good to see some actual numbers and how they compare with the prescribed snow depths. It will also justify your choices of snow depths for the sensitivity simulations.

Minor Comments:

Consistency with the use of Øvstedal/Svartavatn, Vossevangen/Bulken and various basin/station name pairs: There are quite a few places in the figures and the text that the names for the name pairs are used in interchangeably. Would it be possible to keep both names together at all times (i.e. always say "Øvstedal-Svartavatn" together, but not one without the other)?

Abstract line 18-20: Define HBV and NSE.

Line 36: How much that is in € or USD terms? ($\sim$ €3 million, 14 million USD?) Also, it may helpful to quote human casualties as well (if there are any).

Line 65: The meaning here is somewhat unclear here; are you trying to ask if it is the snow or the rain that causes flooding during a rain-on-snow event?

Line 110-111: Given the broad readership of the journal, the authors may have to briefly discuss what the Nash-Sutcliffe Efficiency is. Are NSE = 0.8 and 0.27 good or bad (this is only briefly mentioned much later in the manuscript in section 3.3)?

Line 134: I presume "offline" refers specifically to what happen to the hydrological physics (the hydrological model that is coupled to WRF just respond passively to WRF precipitation and radiative flux forcings). Otherwise, "offline" makes little sense to a reader from the NWP or regional climate modelling community. One way to avoid this problem is to properly define what "offline" means and it only applies to the hydrological model and has nothing to do with WRF.

Figure 1: If possible, add a horizontal legend how big your domain is (like the ones you
commonly see in maps and atlases).

Line 168: "snow observations from NVE" –> "NVE snow observations"

Lines 282-283: Perhaps I am not familiar with the calibration of hydrological models, but this comment about computational costs strikes me as a bit unusual as km-scale WRF should be quite expensive; I would imagine that a month or two to run all those WRF setups that are described in Table 2. It will be educational to put in context how much wall clock time is needed to calibrate the hydrological model for one basin and to do the WRF simulations.

Line 393: "doesn't" –> "does not"

Figures 12, 13: the figures will be easier for the readers to follow if the average heights of the basin are given in the subtitle of each panel.

Please also note the supplement to this comment:
https://www.hydrol-earth-syst-sci-discuss.net/hess-2019-402/hess-2019-402-RC2-supplement.pdf

---

## Author Comment (AC1) · 20 Dec 2019

Review of manuscript "The impact of initial conditions on convection-permitting simulations of flood events " (hess-2019-402) by Li et al.

The manuscript is well written and clearly shows the impact of spinup time and different snow cover on discharge modeling with WRF-Hydro. This work is of high interest for the hydrological modeling community. It should therefore be considered for publication in nhess.

Minor comments: For clarity and conciseness I think the size of the abstract should be reduced to less than 300 words (currently 438 words)

- We have reduced the abstract to less than 300 words.

Page 10, line 285: add units to the values of bias and RMSE. By the way, please consider formulating the bias in percentages instead of mm, in order to facilitate the interpretation of the results.

- Thanks for the comment. We have added the units 'mm' in this sentence: "the Bias and RMSE decrease from 6.74 to 0.56 mm and from 1.2 to 0.55 mm". This correction has been made throughout the manuscript.
  We have also added the bias (%) in percentage in Table 5 for a clearer interpretation. The relevant paragraph is also updated in the revised manuscript:
  "The Nash-Sutcliffe-Efficiency coefficient (NSE) of daily discharge increases from 0.41 to 0.86, while the Bias and RMSE decrease from 5.29 mm (0.88 %) to -0.42 mm (-0.07 %) and from 19.05 mm to 9.03 mm, respectively. …
  Regarding the Bias and RMSE values, they are -0.42 mm (-0.07 %) and 9.03 mm from calibrated WRF-Hydro, 2.52 mm (0.42 %) and 11.3 mm from the upper benchmark, and 7.65 mm (1.27 %) / 2.95 mm (0.49 %) and 18.43 mm / 14.13 mm from two lower benchmarks ($R_{lower/random}$ / $R_{lower/regional}$)."

Page 10, line 289: I guess this "correlation coefficient" comes from the PEST method. Please clarify.

- Yes, the correlation coefficient is from PEST method. We have added the clarification and citation in the Method section: "In addition, the correlation

coefficient matrix of calibrated parameters is also estimated by the PEST method (Doherty, 2015). It tells which two parameters might be linearly dependent (if the correlation coefficient is greater than 0.8)."

Page 11, lines 299-302: this result implies that the runoff coefficient, that is the ratio between discharge and precipitation, is slightly above 1! in order to argue that this is realistic, the authors may emphasize the fact that the case study is in "polar region" and that at the end of October not much ET is expected.

- Thanks for the suggestion. We have added the clarification in the section of 4.1: "ET is small for all the catchments. This is due to the low temperatures at the end of October in western Norway, which lies very close to the Arctic Circle and is dominated by mountainous terrain (Engeland et al., 2004)."

Page 12, line 333: it would be useful to also add the values in %, in order to better assess that the "differences are fairly small".

- Agree. We have added the values of " ±1 %" in the sentence.

Page 12, line 346: "discharge MAE": the whole paragraph is about precipitation so I find it confusing to have this comment about discharge at this place. Is it a typo? Otherwise, this could be moved to the next paragraph.

- We have removed this sentence in order to avoid misunderstanding.

Page 13, line 367: "Unsurprisingly, the snowmelt is due to positive surface temperatures and precipitation" In the Noah LSM, the snowmelt occurs when surface temperature is positive, independently of precipitation. Another land surface model such as NoahMP, which distinguishes between solid and liquid phase in the snowpack would be need to assess the effect of precipitation on snowmelt. So I suggest to remove the comment that the snowmelt is due to precipitation, as the Noah LSM which is used here does not allow to asses that, I think.

- Thanks for the comment. We have now removed this comment.

Page 13, line 374: same comment

- We have corrected it to be: "This is because the limit of melting snow is controlled by the temperature in Noah LSM and a maximum of around 0.5 m snow will be melted away in this case."

Section 5: this section is rather a summary of what has already been explained in the result section, than a discussion. Please consider renaming this section, and also use the past tense.

We have changed this section to be section of "Conclusions" and used the past tense in this section in the revised manuscript.

Typos Page 3, line 87: "in to improve" –> "in order to improve"

- We have corrected it.

Page 15, line 399: "1m" –> "1 m", line 406: "69%" –> "69 %", apply to the entire manuscript

- We have corrected them and applied to the entire revised manuscript.

Page 15, line 429: "performance is depends " –> "performance depends"

- We have corrected it.

Figures

Fig. 6: labels a) and b) are missing

- Labels a) and b) have been added to Figure 6.

Fig 8: Maybe the symbols could be shifted so that they can be all seen?

- We have made a shift so that all the symbols can be seen in the revised Figure 8.

---

## Author Comment (AC2) · 20 Dec 2019

The comments below come from someone from the atmospheric modelling community, and I know some of the co-authors are also atmospheric model users. Hence, some comments may come from a somewhat different angle.

Overall, I find the paper quite easy to read, but there are some issues in the earlier part of the paper that the authors seem to have forgotten readership may include non-hydrologists. It will be nice to put some of the results in the broader context – like the need of high resolution models to deal with complex topography and what the snow-prescribed simulations may mean in climate change.

- Thanks for your suggestions. We have added more detailed contexts and comments in a new Discussion section of the revised manuscript. "Precipitation pattern in Norway varies spatially and is highly affected by the complex topography. More specifically, there is a strong west-east gradient of precipitation, with decreasing amounts as we move eastwards across the mountain range (Dyrrdal, A.V., 2015). To represent the interaction between the atmosphere and the complex terrain realistically there is a need for high spatial resolution in models. For example, for the episode under investigation here Pontoppidan et al. (2017) showed that a 3 km grid scaling represented the precipitation distribution better than an equivalent simulation with 9 km grid spacing, which lacked the observed spatial variability and was unable to show dynamical features like gravity waves. Furthermore, a recent study by Magnusson et al. (2019) found that the grid resolution is important in energy-balanced snow models and the scale error increases with subgrid topographic variability. They also suggested that for snow models the best is to run at the highest possible resolution and any upscaling can bring large regional errors because of model nonlinearities. The results from our study confirmed that convection-permitting simulations fairly fit the requirements of hydrological processes determining flood events in western Norway and address them in a realistic manner.
  Most of the precipitation in Norway is frontal, caused by large-scale cyclone activity in the North Atlantic (Heikkilä et al., 2011). In the West coast region, extreme precipitation occurs in autumn and winter, which is dominated by orography and frontal systems (Dyrrdal, A.V., 2015). In Eastern Norway, where the mountain ranges are located, the annual precipitation is less than the west but with the highest amounts occurring near the steepest surface slopes in winter and fall (Andersen, 1972). In Southeast Norway, however, intense precipitation is dominated by convective precipitation in summer.

Norway, despite its high latitude, has a diverse range of climates including northern Arctic, central alpine and southern maritime and can exhibit an equally wide range of snow regimes (Pall et al., 2019). The role of snowmelt and rainfall is highly relevant for the seasonal flood regimes (Barnett et al., 2005; Vormoor et al., 2016). For example, south-central Norway has an alpine climate, which receives large amounts of precipitation, approximately 30% as snowfall (Saloranta 2014) and has high discharge during spring and early summer due to snowmelt. Southwestern Norway has a maritime climate and the highest precipitation occurs in fall and winter, which often results in flood events (Vormoor et al. 2016). From the results of the study, we can see that the snow feedback to river flow depends on which snow regime the region is in, i.e., little or no snow (CTRL, 0.1 m snow), a lot of snow (1.0 or 2.0 m snow), or somewhere-in-between (0.5 m snow). According to previous studies of the current climate, snow cover above 800m is present for over 200 days of the year in southern Norway (Hanssen-Bauer et al., 2015) and the observed median snow depth of Norway varies from 0.1 - 0.5 m during October - May (1957-2011) (Saloranta, 2012). In some regions of Southern Norway, the snow depth can be up to 2 - 3 m during the late winter (Andreassen and Oerlemans, 2009). Furthermore, Pall et al. (2019) has constructed a rain-on-snow climatology using a 1km gridded observation data (during 1961– 1990) and found that an average monthly count of daily rain-on-snow events varies from 2 to 4 during winter-spring in Southern Norway. Under climate change impact, the snowpack distribution (both temporal and spatial) in Norway will be changing. In general, snowmelt floods will reduce in Norway, while the winter precipitation will increase, which may also lead to larger snow storage, e.g. in mountainous areas in Eastern Norway (Hanssen-Bauer et al. 2015). Meanwhile, other studies also showed that in Norway general increases in both precipitation and temperature (especially warmer winters) will intensificate the risk of rain-on-snow events in certain regions and seasons. Such events can be a major trigger of hazards, i.e., floods and landslides, in the country (Hansen et al. 2014; Pall et al., 2019). The regional pattern of increases and decreases of flood events (both frequency and magnitude) reflects the balance between the different and sometimes counteracting processes, e.g., snowpack dynamics, snowfall vs. rainfall."

Major comment:

1. Be aware who might be reading the paper, and make sure they understand your terminology. Just in the abstract and the Section 1, there are few cases that the authors use acronyms and terminologies that may be readily understandable by hydrologists but not to other readers; proper definitions somehow get mentioned later in the paper, but not on the first incidence that they are used. WRF is a tool commonly used by the NWP and climate community, so it is important to make sure all members of the WRF community to understand what you are talking about. See minor comments for details.

- We agree. Now all the terminologies have been checked and definitions are given at the first time shown, including 'HBV', 'WRF-Hydro', 'WRF', and 'NSE'. Please see them in the revised manuscript and the replies in the 'minor comments' part.

2. As the authors indicate that one needs the high-resolution WRF model to get the precipitation right around the complex topography in their region of interest. One way

to show that is to demonstrate how different the precipitation looks like with the lower- resolution 9km model. I hope it is not too much to ask, but how would Figure 5 looks like for the 9km WRF simulation?

- The 9 km simulation output results in a much coarser precipitation distribution. Figure 1 below shows a comparison of 3 km and 9 km WRF simulation. Regarding how the precipitation distribution is affected by the grid scaling, the details are already addressed in Pontoppidan et. al (2017) where it is shown via a number of metrics that the higher resolution simulations reproduce more faithfully the analyzed event. Our work uses the output of Pontoppidan et al. (2017) as the starting point for the assessment of the hydrological impact. So in this paper, we try to avoid repeating the analyses of Pontoppidan et. al (2017). We have clarified this in the manuscript and emphasized the conclusions.

[Figure]

*Figure 1: (a) Accumulated precipitation during 23.-31. October of CTRL simulation from the 9 km grid spacing; (b) Accumulated precipitation during 23.-31. October from the 3 km grid (which is the original Figure 5 in the manuscript).*

3. Figure 9: Is this due to the short spinup runs have soil dry enough to absorb additional precipitation into the ground during the initial phases of the event? It will be good thing to check within the context of what you see for Figure 7 and will show what the WRF land surface scheme is doing to your results.

- Thank you for your comment. Yes, this is mainly due to the difference of soil moisture. In order to clarify this point, we added an additional figure in the revised manuscript (Figure 8). The figure shows the evolution of basin averaged soil moisture during the period of 23 - 31 October from the different spinup experiments. The soil moisture on the first day clearly differs between spinup times in all catchments. More specifically, the soil moisture on 23 October increases with the increase of spinup time length, which indicates that runs with short spinup have a much drier soil that can absorb additional precipitation during the initial phase of the event, i.e., 23 – 25 October. In general, the soil becomes slightly wetter with increased spinup time. 2 – 3 days after initialization, the soil is saturated no matter the spinup time. This is likely due to the relatively shallow soil depth in the mountainous region of

[Figure]

Southern Norway. We have added this discussion to the revised manuscript.

**Figure 8: The four basin averaged simulated soil moisture with different spinup time, including 1 day (1d-spinup), 3 days (3d-spinup), 5 days (5d-spinup), 12 days (12d-spinup) and 26 days (26d-spinup) for the flooding events during 23 - 31 October, 2014.**

4. If I understood Section 4.5 correctly is that the authors' sensitivity simulations with snow generally falls into two scenarios ("no or nearly-no snow" (control or 0.1 m snow) or "a lot of snow" (0.5+m). In between 0.1 and 0.5m, there will be cases that we may see snow making an impact to the discharges in between those two above binary regimes. This is important in climate change impact – that the snow feedback to river flow will depend on which snow regime we will be in (no snow, a lot of snow, or somewhere-in-between regime). Of course, this is just one event and the authors have given context how rare the event is (50-year return), and it occurred in October (which would have no or little snow).

- This is a valid point and more focused snow-sensitivity study (with many more cases) would be warranted. We have added more references on this topic in the revised manuscript. We have added the following to section 4.5 of the revised manuscript: "From the results, we can see that the snow feedback to river flow depends on which snow regime the region is in, i.e., little or no snow (CTRL, 0.1 m snow), a lot of snow (1.0 or 2.0 m snow), or somewhere-in-between (0.5 m snow). Norway, despite its high latitude, has a diverse range of climates including northern Arctic, central alpine and southern maritime and can exhibit an equally wide range of snow regimes (Pall et al., 2019). In all the role of snowmelt and rainfall is highly relevant for the seasonal flood regimes (Barnett et al., 2005; Vormoor et al., 2016)."

Hence, it will be nice to comment on the following issues: a. Some general comments about the seasonal cycle of extremes in Norway: How often do you see comparable precipitation extreme events during late winter, spring, early summer in which snow would become a factor?

- The precipitation pattern in Norway varies spatially and is highly affected by the complex topography. More specifically, there is a strong west-east gradient of precipitation, with decreasing amounts as we move eastwards across the mountain range. Most of the precipitation is frontal, caused by large-scale cyclone activity in the North Atlantic (Heikkilˈä et al., 2010; Dyrrdal, A.V., 2015). In the western coast region, extreme precipitation occurs in autumn and winter, which are dominated by orographic and frontal precipitation. In Eastern Norway, where the mountain range is located, the annual precipitation is less than the west but with the highest amounts occurring near the steepest surface slope in winter and fall (Andersen, 1972). In Southeast Norway, however, intense precipitation is dominated by convective precipitation in summer.
  As mentioned above, the role of snowmelt and rainfall is highly relevant for the seasonal flood regimes. For example, south-central Norway has an Alpine climate, which receives large amounts of precipitation, approximately 30% as snowfall (Saloranta 2014), and has high discharge during spring and early summer due to snowmelt. Southwestern Norway has a maritime climate and the highest precipitation occurs in fall and winter (Vormoor et al. 2016). Furthermore, Pall et al. (2019) has constructed a rain-on-snow (ROS) climatology using a 1km gridded observation data (during 1961– 1990) and found that an average monthly count of daily ROS events varies from 2 to 4 during winter-spring in Southern Norway.
  We have added the relevant literature and discussion in the revised manuscript.

Are there any reasonable reason to believe the probability of such events during the snow-relevant months to change in the future? I hope this is just some additional literature review. b. What are the typical snow depths that you get by the end of winter? How is that expected to change? I would speculate snow depth would likely to be in "a lot of snow" regime, but it would be good to see some actual numbers and how they compare with the prescribed snow depths. It will also justify your choices of snow depths for the sensitivity simulations.

- Norway is dominated by mountainous regions. As such, ROS events are more severe and complex in Norway compared to other countries. According to previous studies, snow cover above 800m is present for over 200 days of the year in Southern Norway (Hanssen-Bauer et al., 2015) and the observed median snow depth of Norway varies from 0.1 ~ 0.5 m during October – May but with substantial spatial heterogeneity (1957-2011) (Saloranta, 2012). In some regions of Southern Norway, the snow depth can be up to 2 - 3 m during the late winter (Andreassen and Oerlemans, 2009). Studies also show that in Norway, generally, the size of rainfall floods is expected to increase, while meltwater floods will decrease (because of reduced snowpack). Higher temperatures cause the flood time to shift towards earlier spring floods, while the danger of floods in late fall and winter increases (Hanssen-Bauer et al., 2015), which indicates that the risk of ROS events may increase in some regions and seasons (for example Western Norway winter, Northern and Eastern Norway spring), which can be a major trigger of hazards, i.e., floods and landslides, in the country. We have added additional references and discussion in the Discussion section of the revised manuscript.

Minor Comments:

Consistency with the use of Øvstedal/Svartavatn, Vossevangen/Bulken and various basin/station name pairs: There are quite a few places in the figures and the text that the names for the name pairs are used in interchangeably. Would it be possible to keep both names together at all times (i.e. always say "Øvstedal-Svartavatn" together, but not one without the other)?

- We apologize for this lack of clarity. These refer to pairs of stations: the discharge stations are Svartavatn, Bulken, Kinne and Myrkdalsvatn; while the rain gauge stations, which are closest to the basins, are Vossevangen, Mjølfjell, Øvstedal and Myrkdalen, respectively. We have added the nearest rain gauge stations to Table 1 and also clarified this in section 2.3 of the revised manuscript.

Abstract line 18-20: Define HBV and NSE.

- The abstract has been revised. We reduced the number of words of the abstract in response to comments from the first reviewer and added the definitions of all acronyms (e.g., HBV and WRF-Hydro).

Line 36: How much that is in C or USD terms? (~ C3 million, 14 million USD?) Also, it may helpful to quote human casualties as well (if there are any).

- We have added "(~ 16.4 million USD) " in the sentence. While villages were isolated for days, there were no human casualties.

Line 65: The meaning here is somewhat unclear here; are you trying to ask if it is the snow or the rain that causes flooding during a rain-on-snow event?

- We apologize for the lack of clarity. Yes, we were trying to ask which is actually the main cause (rain or snow) of flooding during a rain-on-snow flooding event. But we also feel this is a bit confusing, so we removed the sentence.

Line 110-111: Given the broad readership of the journal, the authors may have to briefly discuss what the Nash-Sutcliffe Efficiency is. Are NSE = 0.8 and 0.27 good or bad (this is only briefly mentioned much later in the manuscript in section 3.3)?

- We have added a clarification in the sentence: " (a perfect model result of NSE is 1.0 and NSE value of 0 indicates that the model predictions are as accurate as the mean of the observed data)".

Line 134: I presume "offline" refers specifically to what happen to the hydrological physics (the hydrological model that is coupled to WRF just respond passively to WRF precipitation and radiative flux forcings). Otherwise, "offline" makes little sense to a reader from the NWP or regional climate modelling community. One way to avoid this problem is to properly define what "offline" means and it only applies to the hydrological model and has nothing to do with WRF.

- Thanks for the comment. We have added the following clarification: "... WRF-Hydro is designed to link across these components and their characteristic scales to provide a modelling framework that can address these

gaps (Gochis et al., 2018). It enables improved simulation of land surface hydrology and energy states and fluxes at high spatial resolution (typically 1 km or less). It can be used in either "offline" (uncoupled to the atmospheric component of the model) or "fully-coupled" modes (the hydrological model components have two-way interactions with the atmospheric component) (Gochis et al., 2015).

…

The work is built upon the study of Pontoppidan et al. (2017) for the simulation of the meteorological processes and the hydrological impact. As such, an 'offline' ('uncoupled') configuration for the WRF-Hydro model is chosen."

Figure 1: If possible, add a horizontal legend how big your domain is (like the ones you commonly see in maps and atlases).

- We have added the grid (i.e., latitude and longitude) in the Figure1. Please see it in the revised manuscript.

Line 168: "snow observations from NVE" –> "NVE snow observations"

- We have corrected it.

Lines 282-283: Perhaps I am not familiar with the calibration of hydrological models, but this comment about computational costs strikes me as a bit unusual as km-scale WRF should be quite expensive; I would imagine that a month or two to run all those WRF setups that are described in Table 2. It will be educational to put in context how much wall clock time is needed to calibrate the hydrological model for one basin and to do the WRF simulations.

- It took ~ 7 mins of wall time to run hydrological part (offline WRF-Hydro) in HPC, comparing with ~13 hours to run WRF simulation during 1 September - 31 October, 2014. The PEST auto-calibration for hydrological model (offline WRF-Hydro model) in Svatavatn basin took about 2 to 3 days of wall time. The WRF-Hydro modelling system has already been widely used in the world and served as one of the core models for national water model in the United States (over 1000 basins in the whole US have been calibrated). In hydrological science, model calibration work is a well-known and had sophsiticated tools, like PEST. So we prefer to not give too much details regarding this. More details can be given from the authors upon request.

Line 393: "doesn't" –> "does not"

- We have corrected it.

Figures 12, 13: the figures will be easier for the readers to follow if the average heights of the basin are given in the subtitle of each panel.

- We have added average elevations of the four basins in the subtitles of Figures 12 and 13 (Figures 13 and 14 in the revised manuscript).

References:

Andersen, P., 1972. The distribution of monthøy precipitation in southern Norway in relation to prevailing H.Johansen weather types. Yearbook for Univ. og Bergen Mat., Naturv. Series, No.1.

Andreassen, L.M. and Oerlemans, J., 2009. Modelling long–term summer and winter balances and the climate sensitivity of storbreen, norway. Geografiska Annaler: Series A, Physical Geography, 91(4), pp.233-251.

Hansen, B.B., Isaksen, K., Benestad, R.E., Kohler, J., Pedersen, Å.Ø., Loe, L.E., Coulson, S.J., Larsen, J.O. and Varpe, Ø., 2014. Warmer and wetter winters: characteristics and implications of an extreme weather event in the High Arctic. Environmental Research Letters, 9(11), p.114021.

Hanssen-Bauer, I., and Coauthors, 2015: Klima i Norge 2100. Norwegian Centre for Climate Services Rep. 2/2015, 204 pp.,

Pall, P., Tallaksen, L.M. and Stordal, F., 2019. A climatology of rain-on-snow events for Norway. Journal of Climate, 32(20), pp.6995-7016.

Saloranta, T.M., 2012. Simulating snow maps for Norway: description and statistical evaluation of the seNorge snow model. The Cryosphere, 6(6), pp.1323-1337.

Dyrrdal, A.V., 2015. Estimating extreme precipitation on different spatial and temporal scales in Norway.